# Dynamic post-translational modification profiling of *Mycobacterium tuberculosis-*infected primary macrophages

Jonathan M Budzik[1,2], Danielle L Swaney[3,4,5], David Jimenez-Morales[3,4,5,6], Jeffrey R Johnson[3,4,5], Nicholas E Garelis[2], Teresa Repasy[2], Allison W Roberts[2], Lauren M Popov[2], Trevor J Parry[2], Dexter Pratt[7], Trey Ideker[7], Nevan J Krogan[3,4,5], Jeffery S Cox[2]*

[1]Department of Medicine, University of California, San Francisco, San Francisco, United States; [2]Department of Molecular and Cell Biology, University of California, Berkeley, Berkeley, United States; [3]Department of Cellular and Molecular Pharmacology, University of California, San Francisco, San Francisco, United States; [4]Quantitative Biosciences Institute, University of California, San Francisco, San Francisco, United States; [5]Gladstone Institutes, San Francisco, United States; [6]Department of Medicine, Division of Cardiovascular Medicine, Stanford University, Stanford, United States; [7]Department of Medicine, University of California, San Diego, La Jolla, United States

*For correspondence:
jeff.cox@berkeley.edu

**Competing interests:** The authors declare that no competing interests exist.

**Abstract** Macrophages are highly plastic cells with critical roles in immunity, cancer, and tissue homeostasis, but how these distinct cellular fates are triggered by environmental cues is poorly understood. To uncover how primary murine macrophages respond to bacterial pathogens, we globally assessed changes in post-translational modifications of proteins during infection with *Mycobacterium tuberculosis*, a notorious intracellular pathogen. We identified hundreds of dynamically regulated phosphorylation and ubiquitylation sites, indicating that dramatic remodeling of multiple host pathways, both expected and unexpected, occurred during infection. Most of these cellular changes were not captured by mRNA profiling, and included activation of ubiquitin-mediated autophagy, an evolutionarily ancient cellular antimicrobial system. This analysis also revealed that a particular autophagy receptor, TAX1BP1, mediates clearance of ubiquitylated *Mtb* and targets bacteria to LC3-positive phagophores. These studies provide a new resource for understanding how macrophages shape their proteome to meet the challenge of infection.

## Introduction

*Mycobacterium tuberculosis* (*Mtb*) is among the most successful human pathogens and forms long-term, chronic infections that can span decades. Macrophages are central to tuberculosis pathogenesis as they are both the major site of *Mtb* replication yet also trigger antimicrobial functions required for host resistance to infection (*Russell, 2011*). The mechanisms by which *Mtb* exploits macrophages and the cell-intrinsic immune effectors that limit *Mtb* replication, as well as how these two properties are balanced during chronic infection, is only partially understood. Uncovering these intimate interactions, which have coevolved over nearly 70,000 years (*Comas et al., 2013*), may reveal novel therapeutic intervention strategies to treat the nearly ten million people who fall ill to tuberculosis infection each year (*World Health Organization, 2018*).

*Mtb* infection of macrophages engages several pattern recognition receptors including toll-like receptor 2 (TLR2) leading to expression of inflammatory mediators (*Gopalakrishnan and Salgame,*

*2016*). After phagocytosis, the bacterial-containing phagosome enters into the canonical endosomal/lysosomal pathway, which is under the control of Rab GTPases, including Rab5 and Rab7. However, the typical maturation of this compartment is actively blocked by *Mtb* and fails to acidify or acquire markers of late endosomes/lysosomes (*Sturgill-Koszycki et al., 1994*), an activity observed in early electron microscopy studies (*Armstrong and Hart, 1971*). While this effect of virulent *Mtb* on host intracellular vesicle trafficking requires the type VII protein secretion system ESX-1 (*MacGurn and Cox, 2007*), the mechanism of phagosomal maturation arrest remains mysterious. Importantly, the ESX-1 system mediates limited perforation of the phagosomal membrane, which in turn activates two cytosolic pathways in host cells (*Watson et al., 2015*; *Watson et al., 2012*; *Manzanillo et al., 2012*). First, activation of the cGAS/STING/TBK1 signal transduction cascade leads to production of type I IFN and a profound antiviral transcriptional response that inhibits host resistance (*Watson et al., 2015*). Second, cytosolic access by *Mtb* also activates ubiquitin-mediated selective autophagy targeting, an evolutionary ancient anti-microbial cellular response that counteracts phagosome maturation arrest by actively targeting microbes to lysosomes (*Via et al., 1997*; *Choy and Roy, 2013*). Although some components of the host autophagic machinery are critical for controlling infection (*Watson et al., 2012*; *Zheng et al., 2009*), it remains unclear whether this antimicrobial effect is dependent on canonical autophagy (*Kimmey et al., 2015*).

Global, unbiased approaches to probe the *Mtb*-macrophage interface have primarily relied on measuring changes in mRNA levels that, while facile, naturally limit the analysis of cellular state toward signal transduction cascades that directly lead to transcriptional outputs (*Ehrt et al., 2001*). While these studies have certainly uncovered insights into the mechanisms of *Mtb* pathogenesis (*Ehrt et al., 2001*; *Berry et al., 2010*), monitoring changes in protein post-translational modifications (PTMs) represents a more comprehensive way to assess changes in cellular state during infection, as essentially all cell biological pathways, including intracellular trafficking, autophagy, nuclear import, and metabolism are regulated by PTMs. Indeed, many intracellular bacterial pathogens hijack or alter normal PTMs of host proteins to manipulate cells and promote their pathogenesis (*Roy and Mukherjee, 2009*; *Patel et al., 2009*; *Fiskin et al., 2016*). However, global PTM analyses require proteomics-based assays that are inherently more difficult than measuring mRNA levels.

Selective autophagy is predominantly regulated by PTMs rather than transcription (*Lamark et al., 2017*), as it is under the control of several types of modification, especially ubiquitylation and phosphorylation (*Herhaus and Dikic, 2015*). Initial targeting of intracellular structures (intracellular pathogens, damaged organelles, protein aggregates, etc.) to autophagy is mediated by ubiquitylation of the cargo. Autophagy receptors then recognize these ubiquitin signals and recruit phagophore membranes to the cargo via interactions with LC3/GABARAP proteins embedded in these vesicles, ultimately enveloping cargo inside a continuous autophagic vacuole competent for fusion with lysosomes. Importantly, PTMs of receptors also govern their activity. For example, ubiquitylation of p62 and phosphorylation of Optineurin increase their activity upon recruitment to cargo, forming a positive feedback loop promoting autophagosome completion (*Peng et al., 2017*; *Matsumoto et al., 2011*; *Heo et al., 2015*). In the case of *Mtb*, some ubiquitin ligases required for selective autophagy have been identified (*Manzanillo et al., 2013*; *Franco et al., 2017*), but their substrates and roles are still mysterious and other ubiquitin ligases are likely required. Likewise, some autophagy receptors have been implicated in autophagy of the *Mtb*-containing vacuole (*Watson et al., 2012*; *Manzanillo et al., 2013*; *Franco et al., 2017*), but many have not been tested (*Rogov et al., 2014*). Thus, global PTM analysis would be likely to reveal components of autophagy involved in *Mtb* infection (*Sarraf et al., 2013*).

Here, we report our findings using mass spectrometry (MS) to comprehensively identify changes in host protein abundance, phosphorylation, and ubiquitylation during a 24 hr time course of primary murine macrophages infected with virulent *Mtb*. These datasets represent a significant resource for future studies and expands upon previously published PTM studies at a single time point in the RAW macrophage cell line (*Penn et al., 2018*) and of tyrosine phosphorylation in primary macrophages (*Sogi et al., 2017*), and has uncovered thousands of significantly modified proteins in response to infection. Bioinformatic analysis indicated that while some PTM changes were enriched in pathways known to be important in *Mtb* pathogenesis (e.g. autophagy), other pathways identified were surprising (e.g. nucleosome assembly). In particular, we identified significant enrichment in the phosphorylation of several autophagy receptors, which guided directed studies that revealed a unique role for TAX1BP1 in cell intrinsic control of *Mtb* infection. Collectively, these findings indicate

that our MS/genetics approach is a powerful way to identify the ways by which macrophages attempt to control intracellular bacterial infection.

## Results

### Proteome-level evaluation of primary macrophage responses to *Mtb* infection

To identify new innate immune pathways modulated during *Mtb* infection, we sought to obtain a deep data set of changes in host protein abundance and post-translational modifications during a time course of macrophage infection. To this end, we infected primary murine bone-marrow-derived macrophages with *Mtb* in biological triplicate, harvested infected cells at 2-, 4-, 6-, 8-, and 24 hr post-infection, and prepared protein lysates (*Figure 1A*). We also performed time-matched mock infections of the same macrophages harvested at an early (0 hr), middle (6 hr), or late (24 hr) time points. Because we identified extremely few changes between uninfected cells at all three time points, we simply matched the *Mtb*-infected cell data from the 2- and 4 hr time points with the data from the 0 hr mock sample, and the 6- and 8 hr time points with the data from the 6 hr mock sample (*Figure 1—figure supplement 1A–C*). Cellular lysate samples were digested with trypsin and a portion of the resulting peptides were used for measurement of protein abundance, with the remaining peptides subjected to separate enrichments using phospho-peptide (*Swaney and Villén, 2016*) or diGly remnant (*Udeshi et al., 2012*) affinity technologies. The diGly-MS approach, however, cannot distinguish ubiquitin modification from modification with either of two different ubiquitin-like proteins, ISG15 and Nedd8, as they leave identical diGly remnants after trypsinization. Thus, although we will refer to diGly remnant peptides as evidence of 'ubiquitylation', it is important to note the possibility that these events represent these rarer modifications. We subjected all peptide samples (abundance, phospho- and diGly-modified peptides) to liquid chromatography-mass spectrometry (LC-MS). Based on the measured *m/z* ratio of the parent ions and fragment ions for each peptide coupled with a database search of the mouse proteome, we determined the amino acid sequence, modification sites, and cognate protein for each peptide. Pair-wise comparison of parent ion MS intensity measurements between technical and biological replicates from the same condition were highly reproducible, as indicated by strong correlation coefficients from global abundance, phospho-enriched, or diGly-enriched samples (*Supplementary file 1*; *Figure 1B*).

To determine the statistically significant changes in response to infection, we analyzed the combined set of biological replicates with the in-house Bioconductor package artMS (*Jimenez-Morales et al., 2019*) to determine the proteins with the most significant changes in abundance and PTMs upon infection ($p < 0.05$ and log2-fold-change $\geq 1$ or $\leq -1$). Importantly, in many cases, we identified significant peptide counts in one experimental condition (e.g. infected) but no corresponding peptides in the cognate control (e.g. uninfected or mock), indicating that these changes were likely biologically relevant, but could not be represented by a mathematical ratio between infected and uninfected due to zero values. To address this issue, we used a method described by Waters et al. in which missing values are imputed by the limit of MS1 detection to calculate an 'imputed value' (*Webb-Robertson et al., 2015*), which allowed us to estimate changes in levels for this category of peptides. The peptides from the global protein abundance or PTM enrichments mapped to thousands of unique proteins (*Figure 1—figure supplement 1D–F*), consistent with deep coverage of the proteome relative to prior *Mtb* proteomics studies (*Hoffmann et al., 2018*). Comparison of peptide levels in *Mtb* infected samples versus uninfected controls revealed over 1000 statistically significant changes in abundance or PTMs during *Mtb* infection (*Figure 1—figure supplements 2–3*; *Figure 2C-G*), indicating pronounced changes to the macrophage proteome.

The number of statistically significant changes in protein abundance or ubiquitylation progressively increased over the time course (*Figure 1C*). In contrast, phosphopeptide changes were bimodal with the greatest number of phosphopeptide changes occurring early (2- or 4 hr post-infection) or late (24 hr) after *Mtb* infection (*Figure 1C*).

Although protein abundance levels are largely determined by mRNA levels under steady state conditions (*Edfors et al., 2016*), environmental changes can result in a transient lack of correlation between transcript and protein levels (*Liu et al., 2016*). However, after cells adapt to their new physiological state, the correlation between mRNA and protein levels is restored (*Liu et al., 2016*). To

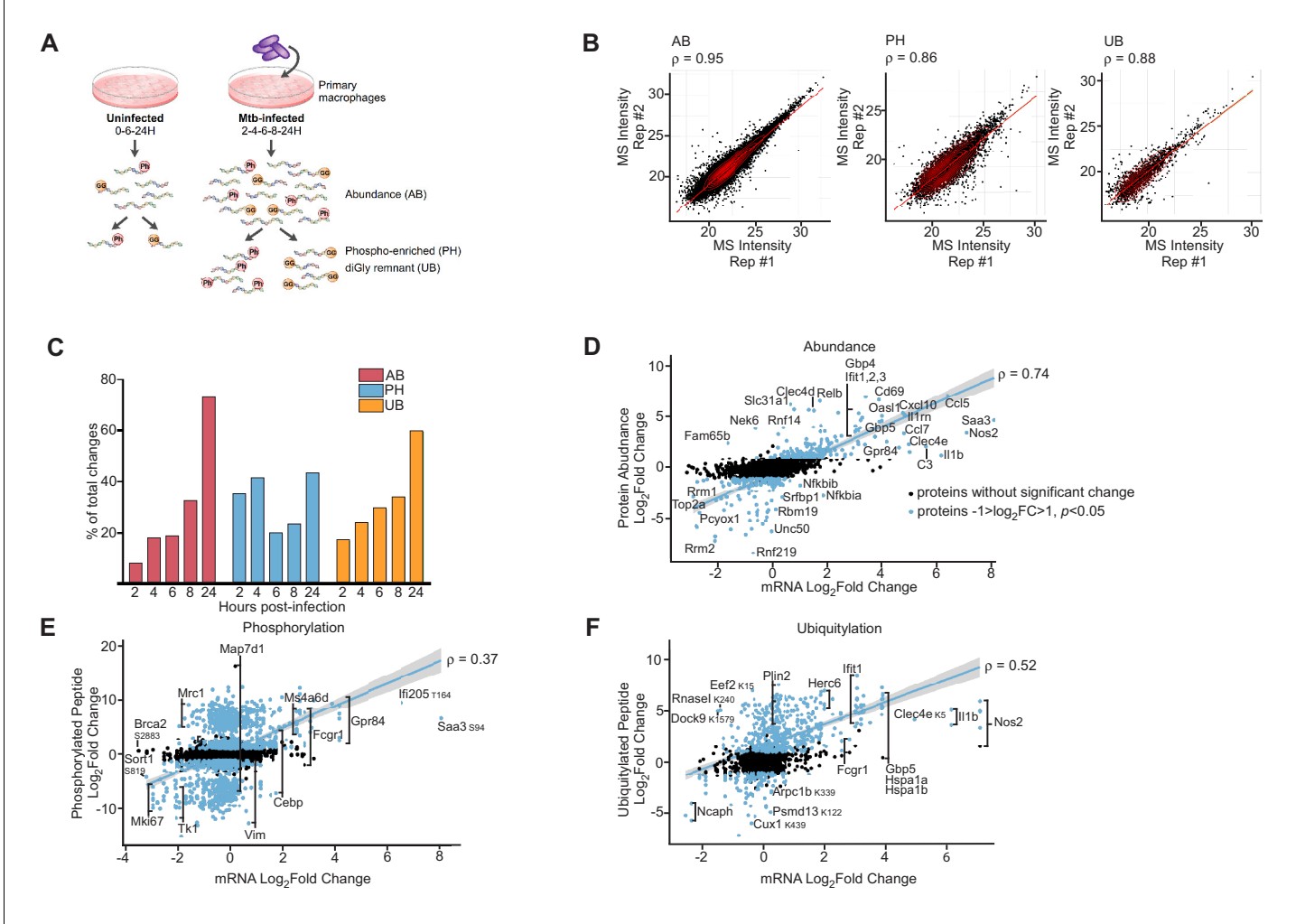

**Figure 1.** Proteomics and RNAseq profiling of *Mtb* infected bone-marrow-derived macrophages. (**A**) Schematic for the experimental design indicating the hours post-infection at which point mock-infected or *Mtb* infected macrophages were harvested and peptides were generated for global protein abundance measurements by LC-MS. Phosphorylated peptides (Ph) or ubiquitylated peptides containing the diGly-remnant (GG) were separately enriched. (**B**) Replica plots of MS intensity measurements for individual peptides in global protein abundance (AB), phosphopeptide (PH), or ubiquitylated peptide (UB) samples. The correlation coefficient is displayed (ρ). (**C**) Percent of total changes (log$_2$(fold change) greater than one or less than −1, p value less than 0.05) occurring at each time point 2–24 hr post-infection for global protein abundance, phosphorylation, or ubiquitylation. (**D–F**) Correlation plot of changes in gene transcription and protein abundance (AB; **D**), phosphorylation (PH; **E**), or ubiquitylation (UB; **F**) at 24 hr post-infection with *Mtb*. Proteins with no statistically significant changes during *Mtb* infection are colored black. Proteins with statistically significant changes during *Mtb* infection are colored blue. The correlation coefficient (ρ) for proteins with statistically significant changes during *Mtb* infection and mRNA levels are shown. Multiple phosphorylation sites were detected in Gpr84, Fcgr1, Ms4a6d, Vim, Map7d1, Mrc1, Tk1, and Mki67 (*Supplementary file 5*). The online version of this article includes the following figure supplement(s) for figure 1:

**Figure supplement 1.** Host protein abundance changes in mock-infected macrophages, and unique host proteins identified at each time point 2–24 hr post-infection with *Mtb* or mock infection.

**Figure supplement 2.** Host protein abundance changes during a time course of *Mtb* infection.

**Figure supplement 3.** Changes in host protein ubiquitylation during a time course of *Mtb* infection.

**Figure supplement 4.** Correlation between host mRNA and protein abundance or PTMs changes during *Mtb* infection.

determine the extent of correlation between the relative changes in RNA and protein abundance after *Mtb* infection, we compared transcriptomics datasets from previous experiments using the same macrophages and *Mtb* strain (*Braverman et al., 2016*) with our proteomic measurements. Overall, our data is consistent with previous findings of dendritic cells treated with LPS (*Berg-Larsen et al., 2013*), as we observed weak correlation between relative protein and RNA abundance

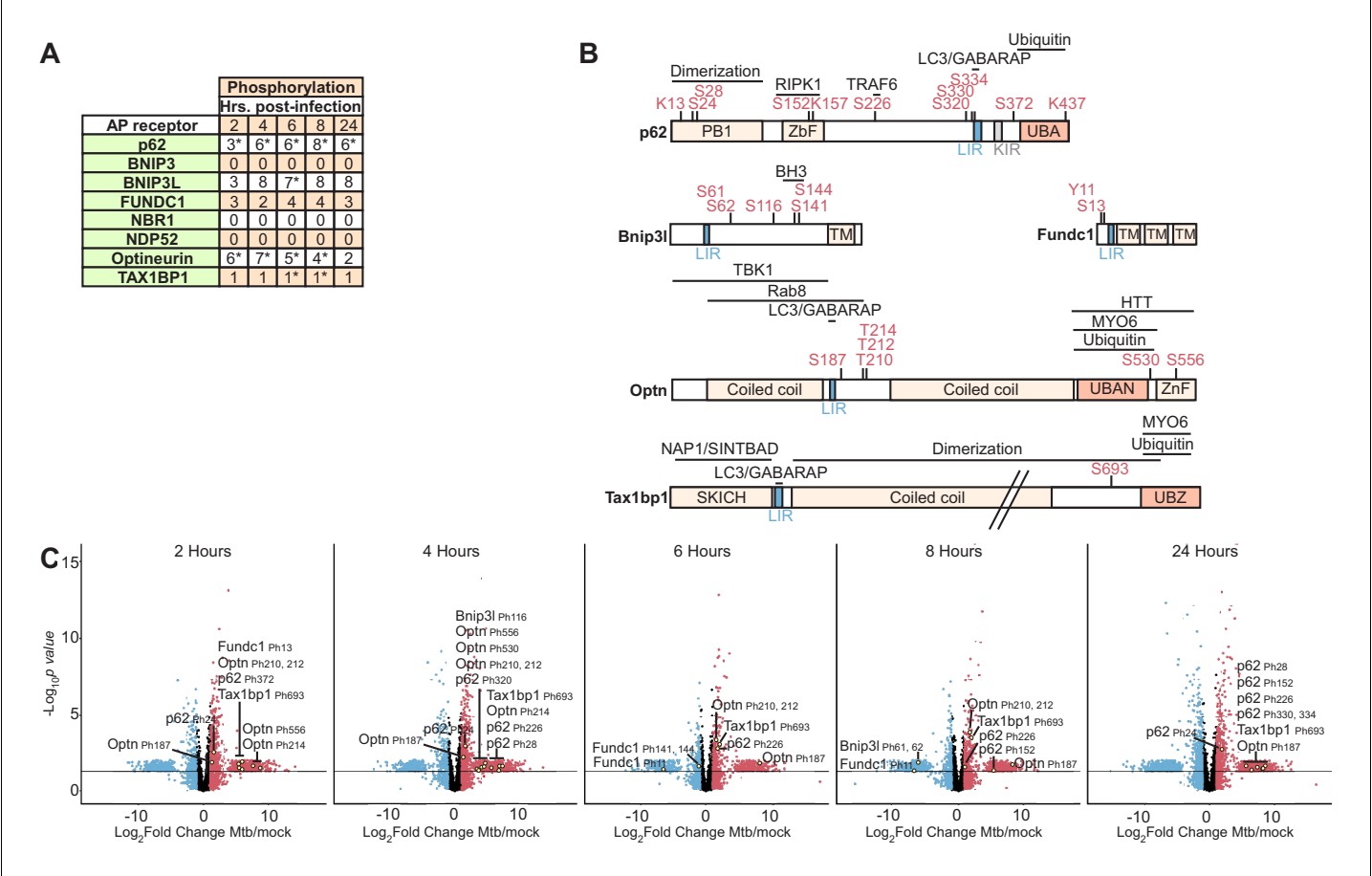

**Figure 2.** Autophagy receptor phosphorylation during *Mtb* infection. (A) Table showing the number of unique phosphosites in each autophagy receptor at 2–24 hr post-infection. * *p* value less than 0.05 and log₂(fold change) greater than one or less than −1 for at least one of the non-imputed phosphosites. (B) Domain organization displaying post-translational modifications in autophagy receptors. The LC3-interacting region (LIR), ubiquitin binding domain (UBAN, UBZ), myosin-6 binding domain (MYO6), SKIP carboxyl homology domain (SKICH), and TANK-binding kinase-1 (TBK1) binding domains are labeled. (C–G) Volcano plots highlighting changes in autophagy receptor phosphorylation at each time point 2–24 hr post-*Mtb* infection. Proteins with a log₂(fold change) greater than one are colored red. Proteins with a log₂(fold change) less than −1 are colored blue. Proteins with a *p* value less than 0.05 (or -log₁₀ (*p* value) greater than 1.3) are above the horizontal black line. Phosphorylated residues and ubiquitylated lysine residues are noted. Statistical analysis of phosphopeptide data was performed with artMS version 1.4.0.

early after infection (6 hr time point, Pearson coefficient ρ = 0.36, *Figure 1—figure supplement 4*), but at 24 hr post-infection the correlation was much stronger (ρ = 0.74, *Figure 1D*) indicating that by this time point the changes in the proteome are largely reflective of changes in the transcriptome. Importantly, this correlation was similar for both non-imputed (ρ = 0.81) and imputed (ρ = 0.78) proteins, indicating that the imputation method accurately captured abundance changes. Interestingly, we observed poor correlations between relative changes in mRNA compared to changes in phosphorylated (ρ = 0.37, *Figure 1E*) or ubiquitylated proteins (ρ = 0.52, *Figure 1F*) at this same late time point. Thus, these results suggest that while changes in RNA abundance is a moderate predictor for changes in protein abundance in response to infection, other biological process controlled by PTMs are at play during macrophage infection, reinforcing the notion that proteomics is a valuable method to identify biological processes during infection that are independent of gene expression.

## *Mtb* infection elicits unique and overlapping changes in protein abundance, phosphorylation, and ubiquitylation

Next, we sought to compare the proteins that changed in abundance, phosphorylation, or ubiquitylation during *Mtb* infection. A complete list of changes in macrophage global protein abundance, phosphopeptide, and diGly-remnant peptides at 2–24 hr post-infection with *Mtb* are found in

*Supplementary files 4–6*. To visualize the differentially modified proteins (global abundance) or peptide (PTMs), we plotted the $\log_2$-fold change in each protein or peptide versus its statistical significance during *Mtb* infection (-$\log_{10}(p$ value); *Figures 2A,3G*, and *Figure 1—figure supplements 2–3*). Non-imputed proteins or peptides with *p* values < 0.05 and a $\log_2$-fold change >1 or<1 were deemed statistically significant (*Figure 3A*). Imputed proteins or peptides were conservatively assigned a *p* value just above the level of statistical significance and thus these values are plotted immediately above the horizontal line demarcating the cut-off *p* value of 0.05 (-$\log_{10}(p$ value)=1.3; *Figure 3A*). Many of the proteins that changed in global abundance and ubiquitylation, but to a lesser degree in their phosphorylation levels, are encoded by interferon-regulated genes such as *Ifit1*, *Ifit2*, *Stat1*, or *Ifih1* (*Figure 3A C*). A comparison of individual proteins that changed significantly in abundance (n = 470), phosphorylation (n = 1489), or ubiquitylation (n = 606) during *Mtb* infection, indicated limited overlap between the three categories (*Figure 3B*).

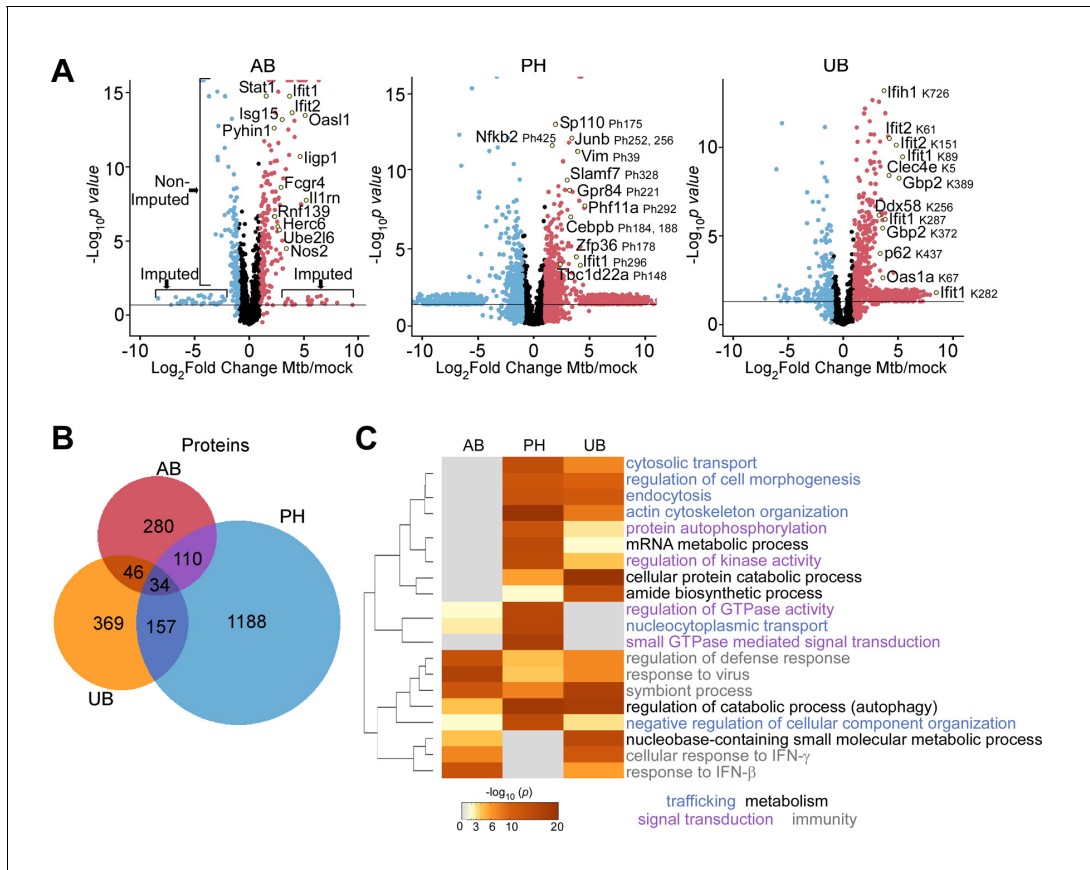

**Figure 3.** Comparison of macrophage proteins changing in abundance, phosphorylation, or ubiquitylation during *Mtb* infection. (A) Volcano plots displaying proteins changing in abundance, phosphorylation, or ubiquitylation at 24 hr post-infection. Proteins with a $\log_2$(fold change) greater than one are colored red. Proteins with a $\log_2$(fold change) less than −1 are colored blue. Proteins with a *p* value less than 0.05 (or -$\log_{10}(p$ value) greater than 1.3) are above the horizontal black line. Phosphorylated residues and ubiquitylated lysine residues are noted. (B) Venn diagram displaying the number of unique and overlapping proteins changing in abundance (AB), phosphorylation (PH), or ubiquitylation (UB) in the aggregate measurements 2–24 hr post-infection. (C) Enriched ontogeny clusters highlighting commonly enriched pathways. Pathways are colored coded into common groups (trafficking, metabolism, signal transduction, immunity). The dendrogram displays clustering of the gene ontogeny pathways.

The online version of this article includes the following figure supplement(s) for figure 3:

**Figure supplement 1.** Changes in host protein phosphorylation during *Mtb* infection.

**Figure supplement 2.** Host protein abundance changes during infection with *Mtb*.

**Figure supplement 3.** Changes in host protein ubiquitylation during *Mtb* infection.

**Figure supplement 4.** Protein kinase and effector complex activity predictions.

**Figure supplement 5.** Overlap of PTMs and host proteins that interact with *Mtb*.

To identify distinct pathways that may be under the control of phosphorylation or ubiquitylation (*Ashburner et al., 2000*), we performed functional pathway enrichment analysis with the bioinformatics pipeline, Metascape (*Tripathi et al., 2015*). Functional pathways involving immunity, cellular trafficking, metabolism, and signal transduction were among the top 20 most statistically significant enriched pathways (*Figure 3C*). As expected based on previous literature, antiviral responses were highly enriched (p=$4.53\times10^{-20}$ in the global abundance dataset; *Figure 3C*), including response to type I (IFN-β p=$4.25\times10^{-16}$, and IFN-α $3.81 \times 10^{-10}$ in the global abundance dataset; *Figure 3C*), likely due to phagosome perforation and activation of the cGAS/STING/TBK1 pathway (*Flynn et al., 1993*; *Cooper et al., 1993*; *Moreira-Teixeira et al., 2018*). Small GTPase mediated signal transduction was also highly enriched in the phosphopeptide datasets, which included Rab GTPases (Rab7, 8b, and 14), guanine nucleotide exchange factors (RabGef1, ArfGef1-2, ArhGef2, Dock1, 2, 5, 8, and 11), and GTPase activators (ArfGap 1, 2, 3, 12, 17, 18, 24, 25, 30, 35) (p=$2.12\times10^{-26}$; *Figure 3C*). As Rab GTPases are involved in endosomal trafficking, and GAPs and GEFs regulate membrane transport, phagocytosis, and control the actin cytoskeleton, enrichment of these functional categories indicates profound reorganization of intracellular trafficking upon infection with *Mtb*. We identified an enrichment in modified proteins in the category 'regulation of catabolic processes' (p=$2.39\times10^{-25}$ in the phosphopeptide dataset; *Figure 3C*), which includes components of the autophagy machinery such as Atg4b, Rubicon, autophagy kinases (Ulk1, Ulk2), and autophagy receptors (Optineurin, Bnip3l). We identified other enriched functional pathways not previously implicated in *Mtb* pathogenesis such as nucleocytoplasmic transport and response to DNA damage (p=$1.92\times10^{-19}$, p=$2.08\times10^{-18}$; *Figure 3C* and *Figure 3—figure supplement 1*). Thus, our unbiased proteomic approach helped elucidate components of pathways known to be important in the host response to *Mtb* and identified new pathways that may play unexpected roles in the polarization of macrophages towards antibacterial defense. Moreover, the functional enrichment analysis highlights the usefulness of measuring changes in the proteome using separate enrichment techniques, allowing one to capture unique host responses to infection that measuring mRNA levels alone does not capture.

## Changes in protein abundance reveal enrichment of antiviral and inflammatory pathways during *M. tuberculosis* infection

The above analysis was performed by aggregation of data over the entire time course of infection. Next, we sought to extract from our data the temporal nature of proteome response by using hierarchical cluster analysis and incorporating the directionality of changes over the individual time points within each of the three data sets (*Figure 2—figure supplements 1–3*). Pathway analysis was subsequently performed on the imputed and non-imputed proteins together (*Figure 2—figure supplements 1–3*). Protein abundance changes clustered into two large groups and several smaller clusters (*Figure 3—figure supplement 2*). The group of proteins that increased in abundance during *Mtb* infection was functionally enriched for inflammatory, antiviral, and 'response to bacterium', suggestive they are predominantly targets of TLR signaling (*i.e.*, TLR2, NOS2, Il1b, FCGR4; *Figure 3—figure supplement 2*; p=$2.19\times10^{-13}$, $2.51 \times 10^{-25}$, $8.32 \times 10^{-15}$). Within the antiviral category, the oligoadenylate synthase-like proteins (OASL1 and OASL2) increased in protein abundance at 8- and 24 hr post-infection, and OAS proteins (OAS1a and OAS2; *Supplementary file 4*) increased at 24 hr post-infection. Strikingly, among all of the non-imputed proteins that changed in abundance, OASL1 had the fourth greatest fold change (*Figure 3A*; 34-fold change, p=$1.56\times10^{-12}$), and there was also a large increase in OASL1 mRNA at the same time point (*Figure 1D*). While OAS proteins activate cleavage of viral nucleic acid (*Choi et al., 2015*), murine OASL1 downregulates the type I IFN response by binding to and inhibiting translation of IRF7 mRNA (*Lee et al., 2013*). OASL1 regulation of the type I IFN response may play a critical role in *Mtb* pathogenesis as OASL1 mRNA was recently described to be increased in macrophages infected with a hypervirulent *Mtb* strain (*Leisching et al., 2017*).

The second cluster of proteins decreased in abundance and included ribosome biogenesis and DNA replication pathways (*Figure 3—figure supplement 2*; p=$8.92\times10^{-10}$, $8.32 \times 10^{-11}$), consistent with studies indicating that *Mtb* infection may arrest macrophage growth (*Cumming et al., 2017*), a strategy utilized by the intracellular pathogen *Legionella pneumophila* (*Moss et al., 2019*; *Sol et al., 2019*). Although the autophagy pathway was not significantly enriched in our abundance

data (*Figure 3—figure supplement 2*), there was a statistically significant increase in the autophagy receptors p62 and TAX1BP1 during *Mtb* infection (*Supplementary file 3*).

## PTM changes during *M. tuberculosis* infection are enriched for immune responses, autophagy, and cellular trafficking

During the 24 hr time course, ubiquitylated substrates clustered into four classes: 1. Peptides that decreased in ubiquitylation during infection, 2. Peptides that had increased ubiquitylation early (2–4 hr) after infection, 3. Peptides that were ubiquitylated between 6–8 hr after infection, and 4. Peptides that were identified as ubiquitylated only at the 24 hr time point (*Figure 3—figure supplement 3*). Autophagy, for example, was enriched at the later time points of infection (class 3 and 4; *Figure 3—figure supplement 3*). We also identified an enrichment in histones among the group of proteins with decreased ubiquitylation during *Mtb* infection (*Figure 3—figure supplement 3*; Class 1, p=$1.82 \times 10^{-7}$, $5.89 \times 10^{-7}$), which is likely reflective of the dramatic changes in the transcriptional program of macrophages when they encounter bacteria (*Zou and Mallampalli, 2014*).

Phosphorylated substrates also clustered together into four discrete classes: 1. Proteins that increased early after infection, 2. Proteins that increased later after infection, 3. Proteins that decreased early after infection, and 4. Proteins that decreased late after infection (*Figure 3—figure supplement 1*). This analysis revealed that actin cytoskeleton organization and small GTPase mediated signal transduction changed dramatically throughout the 24 hr time course of infection as they were significantly enriched in all of the clusters (*Figure 3—figure supplement 1*). Alternatively, several functional categories were only enriched in the early clusters (Classes 1 and 3; cytosolic transport, regulation of IL-2), or in the later clusters (Classes 2 and 4; DNA metabolic process). Notably, enrichment of receptor tyrosine kinase signaling pathways (Class 3) is consistent with the emerging role for Epidermal Growth Factor Receptor in impacting *Mtb* growth within macrophages (*Sogi et al., 2017*; *Stanley et al., 2014*). Additionally, Classes 2 and 4 were enriched for responses to DNA damage and chromosome organization (*Figure 3—figure supplement 1*), which we hypothesize could be secondary to the dramatic transcriptional reprogramming of macrophages and histone ubiquitylation observed at the same time points, as mentioned above (Class 1; *Figure 3—figure supplement 3*). Taken together, this data indicates that phosphorylation mediates prominent rearrangements in cellular trafficking pathways and nucleosomes during the host response to *Mtb* infection.

## Prediction of kinase and ubiquitin ligase activities that may change during *M. tuberculosis* infection

Although the functional significance of the phosphorylation changes we observed during *Mtb* infection await determination, we sought to use our phosphoproteomic data to predict kinases that may be activated during *Mtb* infection. To this end, we used the PhosFate bioinformatic tool to compare the differentially phosphorylated substrates we identified during *Mtb* infection with publicly available datasets that encompass phosphorylation changes determined under a wide variety of experimental perturbations (*Ochoa et al., 2016*). Correlations identified in this way can be used to infer kinase regulation from proteomic data and lead to defined hypotheses that can be tested directly. Importantly, this approach identified kinases expected to be activated during *Mtb* infection, including those involved in inflammatory (IKBKB) and stress responses (several MAP kinases) as well as type I IFN signaling (JAK/STATs). Conversely, we also noted potential decreased activity of several cyclin dependent kinases, consistent with the finding that primary macrophages stop proliferating upon infection (*Cumming et al., 2017*; *Figure 3—figure supplement 4A*). Interestingly, several unexpected kinases were identified at the earliest timepoints, including GTF2F1 and Aurora, suggesting that these may regulate previously unrecognized pathways that underlie macrophage responses to infection. GTF2F1 interacts with c-Jun N-terminal kinases (JNK) 1α1 and 3α1, known for their role in cellular stress responses (*Chen et al., 2014*). Aurora kinase along with ATR, PLK1, Aurora kinase A, WEE family kinase proteins (WEE1 and PKMYT1), and CDK1, a kinase cascade that respond to damaged DNA and affects mitosis progression at the G2-M checkpoint (*Willems et al., 2018*; *Joukov and De Nicolo, 2018*; *Schmidt et al., 2017*). Indeed, persistent TLR2-mediated activation of ATR has recently been found to promote polyploid macrophage differentiation, a feature of granulomas, indicating that this kinase cascade might play an important role in development of chronic

infection (*Herrtwich et al., 2018*). Finally, this kinase prediction analysis also revealed potential modulation of the ROCK/LIMK1/LIMK2 kinase cascade that regulates actin filament dynamics, an effect that has not been described during *Mtb* infection (*Figure 3—figure supplement 4A*; *Ohashi et al., 2000*; *Sumi et al., 2001*). Intriguingly, major disorganization of actin was reported during infection of macrophages with a related pathogen, *Mycobacterium avium*, *Guérin and de Chastellier (2000)* suggesting that virulent mycobacteria may exploit changes in actin filament dynamics to infect host cells.

To predict potential ubiquitin ligases involved in host responses to *Mtb* infection, we used Phos-Fate to predict activity changes in kinases with known protein-protein interactions, and then manually searched this list of kinase complexes for those containing ubiquitin ligases as a component of the complex (*Figure 3—figure supplement 4B*). In addition to revealing the TRAF-2 and TRAF-6 ubiquitin ligase complexes, which have known roles in immune signaling during infection (*Rahman et al., 2014*), this analysis identified ubiquitin ligase complexes, including CUL3, RBX1, and MDM2, that have not been previously implicated in *Mtb* pathogenesis (*Figure 3—figure supplement 4B*). Comparison of our current PTM data with our *Mtb*-human protein-protein interaction cell network (*Penn et al., 2018*) revealed that 47 of the 187 host interactors were differentially modified in mouse cells, 15 of which were also identified in similar PTM profiling experiments during *Mtb* infection of the RAW 264.7 macrophage-like cell line (*Figure 3—figure supplement 5*). Taken together, these bioinformatic analyses are valuable for maximizing the usefulness of our proteomics resource by providing a way to prioritize subsequent experiments to guide downstream mechanistic studies.

## p62, NBR1, optineurin, and TAX1BP1 colocalize with *M. tuberculosis* during macrophage infection

The autophagy pathway was prominently enriched in both PTM datasets (*Figure 3C*, *Figure 3—figure supplement 3*), with a significant number of phosphorylation events that occurred on autophagy cargo receptors (*Figure 2A*). While both p62 and NDP52 have been linked to *Mtb* targeting to the autophagosome (*Watson et al., 2012*), we found that only p62 was modified. In fact, among the autophagy receptors, p62 had the greatest number of phosphorylated residues that changed during *Mtb* infection and was among the most highly modified proteins in our proteomics datasets. We identified many of the previously recognized p62 phosphosites, including Ser-152 in the RIPK1 domain and Ser-226 in the TRAF6-binding domain (*Richter et al., 2016*), but we have identified novel residues as well (Ser-28, Ser-320, Ser-372; *Figure 2B*). In an effort to identify additional autophagy receptors that participate in targeting of *Mtb*, we examined six other members of this family of proteins for PTM changes, which are hallmarks of activation (*Grumati and Dikic, 2018*). Four receptors (BNIP3L, Optineurin, FUNDC1, and TAX1BP1) were selectively modified upon *Mtb* infection; these modifications occurred in functional domains of these receptors (*Figure 2B*) and changed through the time course of infection (*Figure 2A,C–G*).

To begin to test the functional role of these six autophagy receptors (BNIP3, BNIP3L, FUNDC1, NBR1, Optineurin, and TAX1BP1) during *Mtb* infection, we sought to determine if any localize to the bacterial-containing phagosome. We engineered FLAG-tagged constructs of all six receptors and individually transduced RAW macrophages with lentiviral expression vectors for each construct, as well as a similar p62 construct as a positive control and an untransduced negative control (*Figure 4—figure supplement 1A and B*). We infected these cells with mCherry-expressing *Mtb* and visualized colocalization with the receptors via confocal immunofluorescence microscopy using antibodies that recognize the FLAG epitope (*Figure 4A*). Importantly, we detected colocalization of *Mtb* with NBR1, Optineurin, TAX1BP1 and p62 (*Figure 4A*). In contrast, while we observed specific staining in cells that expressed tagged BNIP3, BNIP3L, or FUNDC1, these structures did not coincide with the *Mtb* phagosome to any appreciable amount. Quantification of the percent of *Mtb*-containing phagosomes that colocalized with each receptor revealed that both NBR1 and TAX1BP1 localized to a similar extent as p62, whereas colocalization with Optineurin was considerably less frequent. The ~30% colocalization rate of *Mtb* with p62 is consistent with previous findings (*Watson et al., 2012*). Together, these results revealed that p62, Optineurin, and TAX1BP1, are phosphorylated during *Mtb* infection and also colocalize with *Mtb*.

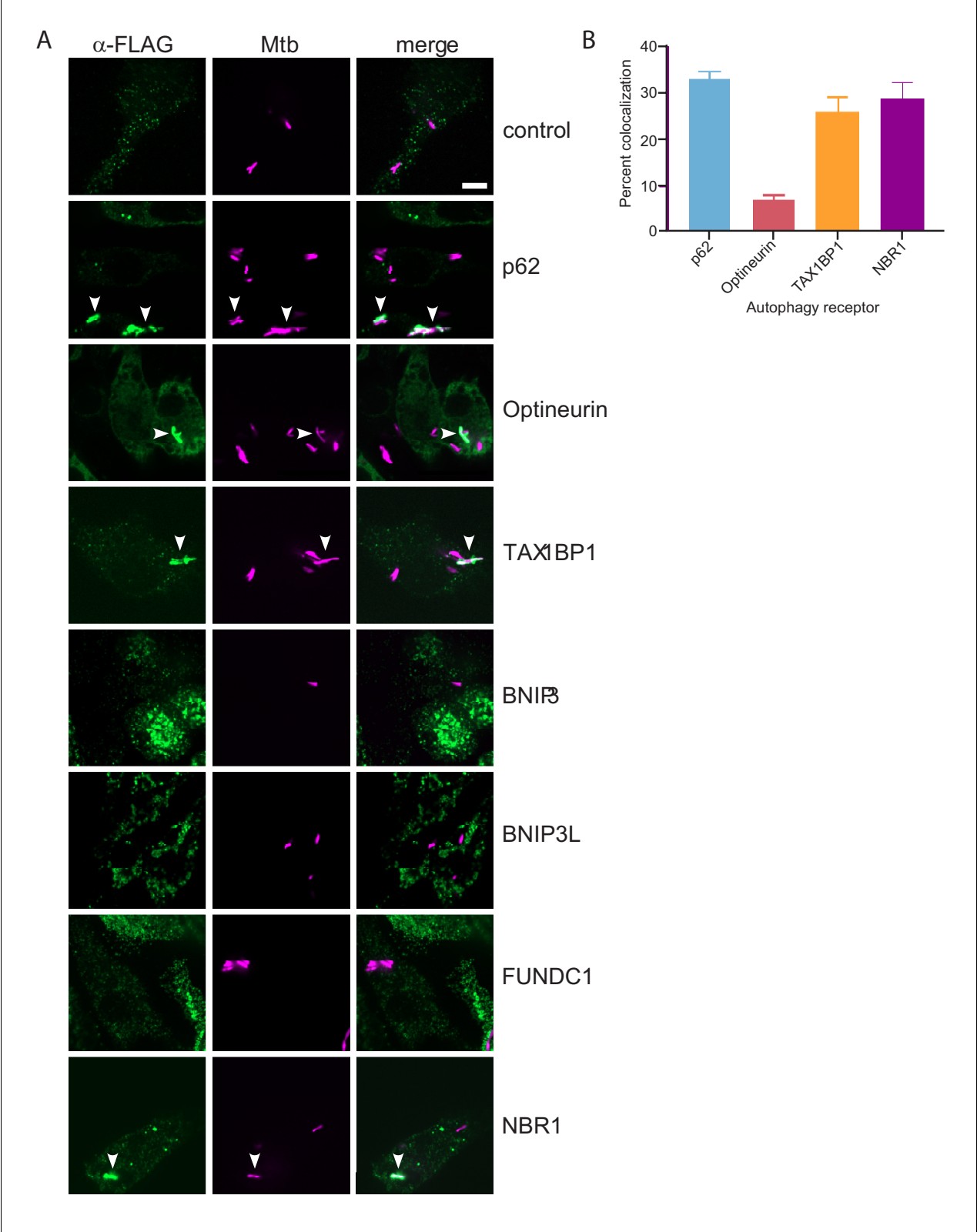

**Figure 4.** TAX1BP1, p62, optineurin, and NBR1 colocalize with *Mtb*. (**A**) Fluorescence images of RAW macrophages expressing FLAG-tagged autophagy receptors (green) with mCherry-expressing *Mtb* (magenta). White bar denotes 10 μm. (**B**) Quantitative analysis of *Mtb* colocalization with FLAG-tagged autophagy receptors. Results are the ± SEM for three technical replicates. Colocalization of mCherry-expressing *Mtb* with the FLAG-
*Figure 4 continued on next page*

*Figure 4 continued*

tagged autophagy receptors was 33 ± 1.7% for p62, 29% ± 3.3% for NBR1, 7% ± 1% for Optineurin, and 26 ± 3.2% for TAXBP1. There were not significant amounts of colocalization with *Mtb* in macrophages expressing FLAG-tagged BNIP3, BNIP3L, or FUNDC1.

The online version of this article includes the following figure supplement(s) for figure 4:

**Figure supplement 1.** Epitope tagged autophagy receptors were expressed in RAW macrophages.

## TAX1BP1 deficiency results in accumulation of ubiquitylated *M. tuberculosis*

We decided to further investigate TAX1BP1 because it colocalized strongly with *Mtb*, has not been previously implicated in *Mtb* infection, and has a unique domain architecture (*Figure 4A*). TAX1BP1 contains a LC3-interacting region (LIR) that binds to LC3 and GABARAP that are common in autophagy receptors (*Whang et al., 2017*). It also includes several other functional domains including a N-terminal NAP1/SINTBAD interaction domain (*Fu et al., 2018*), a dimerization domain, TRAF6 interaction domain (*Ling and Goeddel, 2000*), and overlapping C-terminal ubiquitin and myosin-6 binding domains (*Morriswood et al., 2007*; *Ceregido et al., 2014*; *Figure 2B*). To evaluate the contribution of TAX1BP1 in ubiquitin-mediated selective autophagy of *Mtb*, we used a genetic approach in combination with high throughput immunofluorescence microscopy and automated colocalization analysis from images collected from over 1000 infected cells. First, we quantified ubiquitin, p62, and phospho-TBK1 recruitment to mCherry-expressing *Mtb* in wild-type, $p62^{-/-}$, or $Tax1bp1^{-/-}$ targeted knockout primary bone-marrow-derived macrophages (*Figure 5A*). Similar to prior results showing that p62-deficient macrophages are impaired in the targeting of *Shigella* vacuolar remnants to ubiquitin (*Dupont et al., 2009*), we found decreased targeting of *Mtb* to ubiquitin (FK2) in $p62^{-/-}$ macrophages compared to wild-type cells, consistent with an early role for p62 in signal amplification required for full cargo ubiquitylation (*Figure 5B*; *Peng et al., 2017*). In contrast, infected $Tax1bp1^{-/-}$ cells gave rise to significant increases in *Mtb* colocalization with ubiquitin, phospho-TBK1, and p62 compared to infected wild-type macrophages (*Figure 5B*), indicating that this receptor functions subsequent to p62 and the assembly of signaling complexes to the phagosome, similar to that observed with *Salmonella eneterica* (*Tumbarello et al., 2015*). As an alternative genetic approach, we generated Cas9-expressing bone-marrow-derived macrophages and transduced them with lentiviruses that expressed guide RNAs targeting *Tax1bp1* exon 9 or 10. In pooled populations of edited macrophages, the editing efficiency at exon 9 or 10 calculated by TIDE analysis (*Brinkman et al., 2014*) was 90.0% and 84.8%, respectively. In concordance with our observations in cells harvested from $Tax1bp1^{-/-}$ knockout mice, infected *Tax1bp1* edited macrophages displayed increased *Mtb* colocalization with ubiquitin, phospho-TBK1, and p62 compared to macrophages transduced with scramble control guides (*Figure 5—figure supplement 1*).

The increased accumulation of ubiquitin and p62 surrounding *Mtb* in $Tax1bp1^{-/-}$ cells strongly suggested that the autophagic pathway was blocked at a step downstream of ubiquitin deposition and recognition by initial ubiquitin-binding receptors, but prior to delivery to lysosomes. By interacting with myosin VI and Tom1, TAX1BP1 is thought to recruit endosome-derived vesicles to *S. typhimurium* to promote the fusion step with lysosomes (*Tumbarello et al., 2012*). Indeed, in these studies inhibition of myosin VI or Tom1 blocked delivery of the bacteria to lysosomes and led to increased accumulation of processed LC3 on the autophagosome (*Tumbarello et al., 2012*). However, during *Mtb* infection of *Tax1bp1* edited or targeted knockout macrophages, we observed a slight yet significant decrease in colocalization with LC3 (*Figure 6A and B*) that persisted in the presence of interferon-γ (IFN-γ) stimulation (*Figure 6—figure supplement 1*). Taken together, these results indicate that TAX1BP1 functions in a specific step of autophagosomal targeting and is required for efficient delivery LC3-containing autophagosomal membranes to the *Mtb* containing phagosome.

## TAX1BP1 deficiency augments *M. tuberculosis* growth

To test if TAX1BP1 mediates cell-intrinsic resistance to *Mtb* growth, we infected bone-marrow-derived macrophages with wild-type *Mtb* at a multiplicity of infection of 1 and enumerated colony-forming units (CFUs) on days 0, 3, and 5 post-infection. We observed a moderate yet statistically significant increase in *Mtb* CFUs at 3- and 5 days post-infection in $Tax1bp1^{-/-}$ macrophages compared

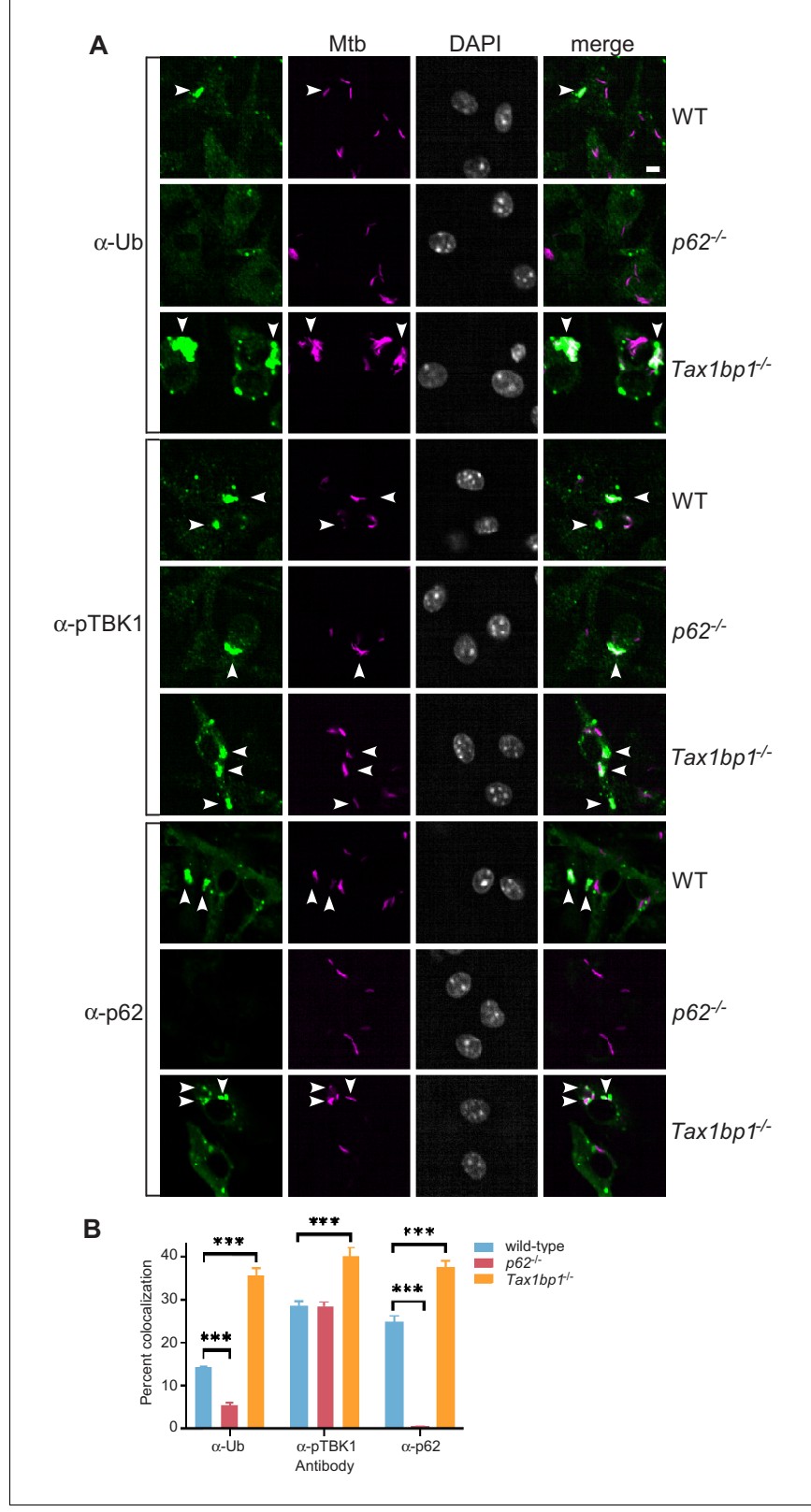

**Figure 5.** Accumulation of *Mtb* targeted to ubiquitin (FK2), p62, and phospho-TBK1 in TAX1BP1 targeted knockout macrophages. (**A**) Fluorescence images of wild-type, *p62⁻/⁻*, or *Tax1bp1⁻/⁻* bone marrow-derived macrophages infected with mCherry-expressing *Mtb* (magenta) and immunostained with antibodies to ubiquitin (FK2), p62, or phospho-TBK1 (green). Nuclei were stained with DAPI (white). White bar denotes 10 μm. (**B**) *Figure 5 continued on next page*

*Figure 5 continued*

Quantitative analysis of *Mtb* colocalization with ubiquitin, p62, and phospho-TBK1 at 8 hr post-infection. Results are the means ± SEM from at least four technical replicates and a single experiment representative of three independent experiments. *** p value less than 0.001 by *t*-test.

The online version of this article includes the following figure supplement(s) for figure 5:

**Figure supplement 1.** Accumulation of *Mtb* targeted to ubiquitin (FK2), p62, and phospho-TBK1 in TAX1BP1 edited macrophages.

to wild-type controls (*Figure 6C*). Likewise, using a bioluminescent *Mtb* reporter strain (*Roberts et al., 2019*) at a multiplicity of infection of 2 or 5, bacterial growth was similarly accelerated in *Tax1bp1⁻/⁻* macrophages (*Figure 6D and E*; *Penn et al., 2018*). In contrast, *Tax1bp1⁻/⁻* macrophages were still capable of restricting *Mtb* growth when stimulated with IFN-γ, an effect likely due to production of normal levels of nitric oxide, the major anti-*Mtb* effector of murine macrophages induced by this cytokine (*Figure 6E*; *Figure 6—figure supplement 1*). These results reveal that, in the absence of IFN-γ, TAX1BP1 deficiency leads to increased *Mtb* growth in macrophages ex vivo. Although TAX1BP1 regulates NF-κB-mediated inflammation in some contexts (*Iha et al., 2008*), levels of secreted pro-inflammatory cytokines IL-6, IL-12, and TNF-α did not significantly change in these experiments (*Figure 6—figure supplement 2*). However, we measured a modest yet significant increase in the production of type I interferon, a cytokine that promotes bacterial growth in vivo, consistent with previous reports of TAX1BP1 action (*Parvatiyar et al., 2010*).

## Discussion

In this study, we generated a comprehensive database of changes in protein abundance, phosphorylation, and ubiquitylation during *Mtb* infection of primary bone marrow-derived macrophages. Measuring host global PTM responses represents a powerful way to uncover novel innate immune pathways and the pathogenic strategies of infectious microbes. Indeed, such global strategies have already begun to elucidate mechanisms of viral pathogens (*Faust et al., 2018*; *Albin et al., 2013*). We envision that the thousands of PTM changes we have identified represents an ensemble of the various pathways that are integrated to meet the challenge of infection, but also reflect the pathogenic strategies of *Mtb*. Several of the functional pathways identified in our pathway analysis of proteomic changes have been previously described to play an important role in host immune responses to *Mtb*. These functional pathways include inflammatory responses (*Toossi, 2000*), antiviral response (*Moreira-Teixeira et al., 2018*), autophagy (*Vergne et al., 2006*), receptor tyrosine kinase signaling (*Sogi et al., 2017*), actin cytoskeleton (*Guérin and de Chastellier, 2000*), endosomal transport and GTPases (*Pei et al., 2012*). In addition to these host pathways, our analysis also revealed changes in macrophage functional pathways not previously known to play a role during infection such as mRNA metabolism and DNA replication, highlighting potential areas of new biology involved in the host response to *Mtb*. Using both host and bacterial genetics will help deconvolve the various changes in PTMs that are specific to individual pathogens, those that are more general, and those that occur under many different stresses.

In particular, our analysis revealed new information about the temporal dynamics of autophagy receptor phosphorylation and identified previously unrecognized PTMs in autophagy receptor functional domains. In contrast to the bimodal peak in overall host protein phosphorylation early (2–4 hr) and late (24 hr) after infection, autophagy receptor phosphorylation, including TAX1BP1, was greatest in between these two peaks (6–8 hr). By probing for PTM changes across time points until the first signs of macrophage monolayer disruption enabled us to capture the dynamics of these transient modifications, some of which may be reflective of the changing conditions within the *Mtb* phagosomal niche (*Rohde et al., 2007*).

Based on these significant dynamics of the host response over the time course of infection in the absence of pre-stimulation with cytokines (e.g. interferon-γ), we similarly suspect dramatic global changes in protein abundance and post-translational modifications following macrophage activation. Indeed, phosphoproteomic analysis of phagosomes from uninfected IFN-γ stimulated macrophages identified changes in many cellular protein networks that overlap with our *Mtb* proteomics analysis including trafficking, cytoskeleton, and immune responses (*Trost et al., 2009*). Furthermore, RNAseq

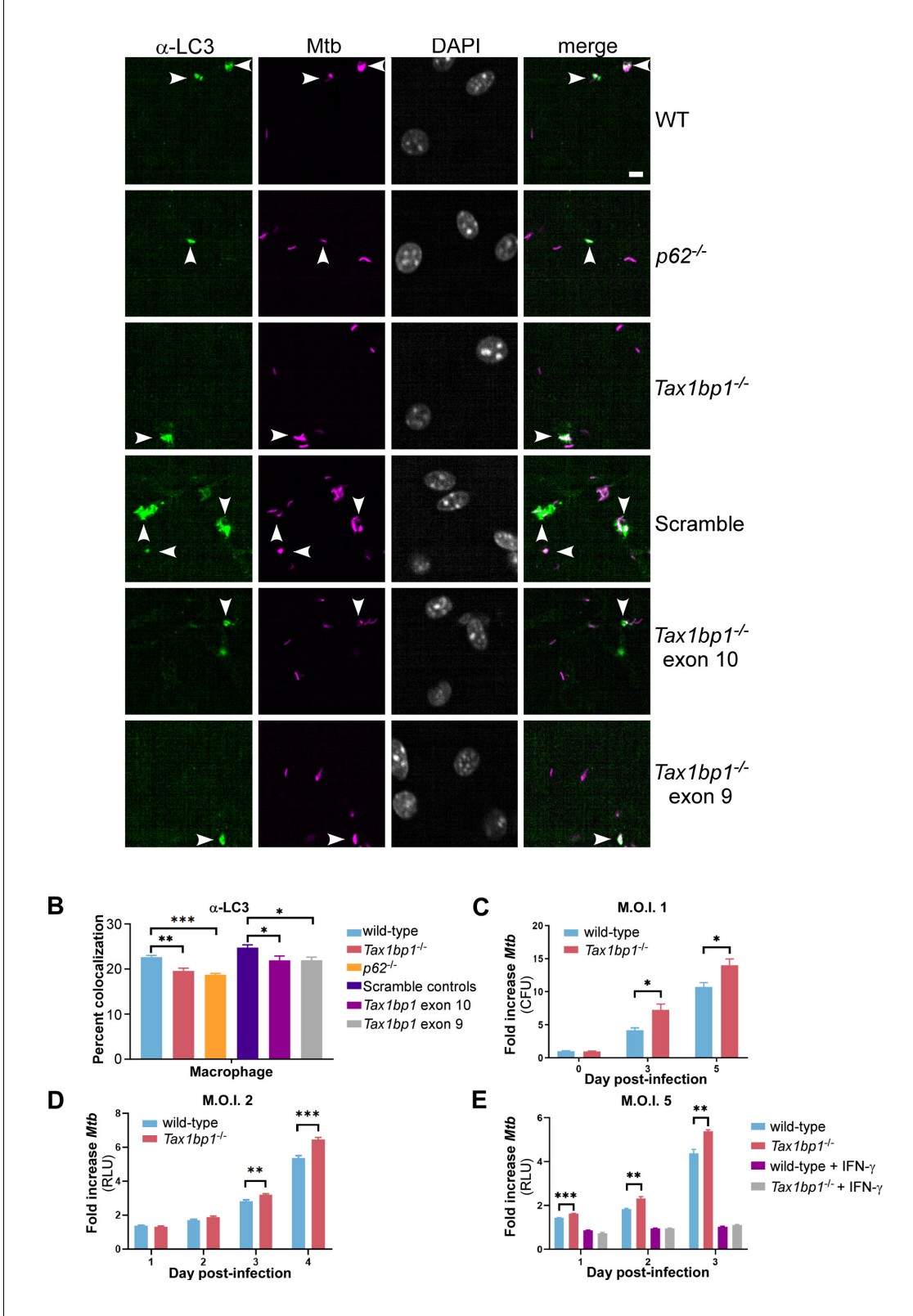

**Figure 6.** TAX1BP1 deficiency abrogates targeting of *Mtb* to LC3 and enhances the growth of *Mtb* in macrophages ex vivo. (**A**) Fluorescence images of bone-marrow-derived macrophages infected with mCherry-expressing *Mtb* (magenta) and immunostained with α-LC3 antisera (green). Nuclei were stained with DAPI (white). White bar denotes 10 µm. (**B**) Quantitative analysis of *Mtb* colocalization with LC3 at 8 hr post-infection. Results are the means ± SEM for four technical replicates of a single experiment representative of two independent experiments. * p value less than 0.05 by *t*-test. ** p

*Figure 6 continued on next page*

*Figure 6 continued*

value less than 0.01. *** p value less than 0.001. (C) Bone-marrow-derived macrophages infected with *Mtb* at a multiplicity of infection of 1 were enumerated for CFU at days 0, 3, and 5 post-infection. Fold change in CFUs is displayed. Results are the means ± SEM of three technical replicates. (D and E) Luminescence from macrophages infected with *Mtb* harboring *luxBCADE* at a multiplicity of infection of 2 (D) or 5 (E) was measured daily. Fold change in relative light units (RLUs) is displayed. * p value less than 0.05 by *t*-test. ** p value less than 0.01. *** p value less than 0.001. Results are the means ± SEM of five technical replicates of two independent experiments.

The online version of this article includes the following figure supplement(s) for figure 6:

**Figure supplement 1.** IFN-γ stimulation abrogates targeting of *Mtb* to LC3.
**Figure supplement 2.** Nitrite and cytokine measurements from supernatants of macrophages infected with *Mtb*.

---

revealed a profound change in the transcriptional profile of *Mtb*-infected macrophages pre-stimulated with IFN-γ (*Braverman et al., 2016*), which also included immune metabolic reprogramming. Since there is limited overlap between PTM and mRNA changes, together this suggests proteomic profiling of *Mtb*-infected macrophages pre-stimulated with IFN-γ may shed light on additional mechanisms enabling IFN-γ-mediated control of infection.

We found *Mtb* infection elicited post-translational modifications in several autophagy receptors including dephosphorylation of two residues in the BH3 domain of BNIP3L (*Figure 2B E*), an atypical BH3-only protein involved in mitochondrial clearance (*Novak et al., 2010*) and cell death (*Imazu et al., 1999*). We are not aware of previous reports describing phosphorylation in its BH3 domain residues Ser-141 and Ser-144 (*Figure 2B E*). Our phosphoproteomic analysis also identified new phosphosites in FUNDC1 (Tyr-11) and Optineurin. While affinity purification of GFP-tagged optineurin previously revealed TBK1-mediated phosphorylation at residues adjacent to the LIR and UBAN domains of Optineurin (*Richter et al., 2016*), we found two novel phosphosites between its LIR and coiled coil domains at Thr-210 and Thr-212, in addition to a phosphosite at Ser-556 in its zinc finger domain (*Figure 2B–G*). Since several of these phosphorylated residues occur in important domains of the autophagy receptors, we predict that they have important functional consequences in host response to *Mtb* and likely other infectious agents.

The autophagy receptor p62 was also phosphorylated in several functional domains during *Mtb* infection. Several of these phosphosites were also identified in IFN-γ stimulated macrophages (Ser-334) (*Trost et al., 2009*), mouse tissue (Ser-24, Ser-330) (*Huttlin et al., 2010*), or found to be phosphorylated by TBK1 (Ser-152, Ser-226) (*Richter et al., 2016*). However, the three phosphosites we identified during *Mtb* infection at Ser-28, Ser-320, and Ser-372 are, to our knowledge, novel and may play a role in cargo selectivity.

In addition to undergoing extensive phosphorylation, p62 protein abundance and ubiquitylation increased during *Mtb* infection (*Supplementary file 4*). While ubiquitylation of human p62 at Lys-157 has been previously reported in HEK293T (*Peng et al., 2017*) and SUMOylation of human p62 at Lys-437 was discovered in HeLa and U2OS cell lines (*Hendriks et al., 2017*), we discovered a third lysine residue ubiquitylated in the dimerization domain during *Mtb* infection (Lys-13) which has not been described previously. Intriguingly, while Lys-7 ubiquitylation by TRIM21 is critical for p62 oligomerization and protein sequestration (*Pan et al., 2016*), a K13R mutant was still ubiquitylated in HEK293T cells (*Pan et al., 2016*), indicating that modification at Lys-13 may play a unique role during infection. Recent studies suggest that ubiquitylation of p62 destabilizes its UBA domain and disrupts dimerization of the UBA, thereby activating its recognition of poly-ubiquitin chains (*Peng et al., 2017*). Indeed, we found that p62-deficient macrophages were unable to recruit ubiquitin to *Mtb*, supporting a model whereby p62 recruitment leads to a feed forward loop with further recruitment of ubiquitin to the autophagosome (*Figure 7*).

Finally, the autophagy receptor, TAX1BP1, was phosphorylated during *Mtb* infection, and its protein abundance also increased (*Figure 2B*; *Supplementary file 4*). In contrast to p62's function in recruiting ubiquitin to the *Mtb* autophagosome, TAX1BP1 appears to function at a step downstream of p62, ubiquitin, and phospho-TBK1 recruitment, enabling full maturation of the ubiquitylated *Mtb* phagosome to a LC3-positive phagophore (*Figure 7*). Intriguingly, TAX1BP1 is not required for global autophagy in T cells or MEFs (*Whang et al., 2017*), indicating that TAX1BP1 may play a role in autophagy of select cargos. However, TAX1BP1 has been implicated as an autophagy receptor important for clearance of *Salmonella* in a manner that requires myosin VI, one of TAX1BP1's binding partners (*Tumbarello et al., 2015*). Likewise, the fact that

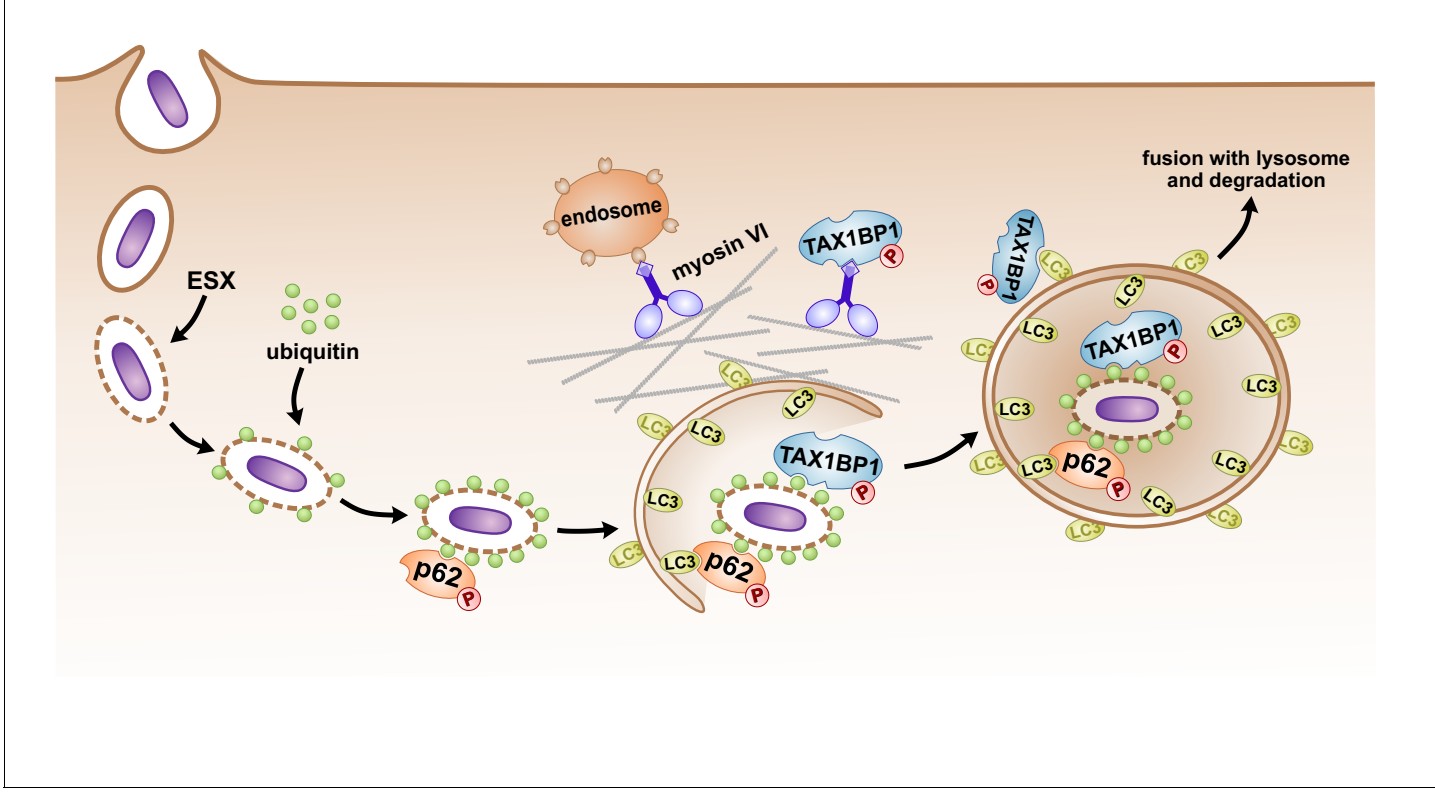

**Figure 7.** Model of autophagosomal targeting *Mtb* by sequential recruitment of autophagy receptors and their binding proteins. Initial recruitment of p62 to the nascent autophagosome leads to amplification of ubiquitin recruitment to *Mtb*. Subsequently in a step downstream of p62 recruitment, TAX1BP1 mediates targeting of ubiquitylated *Mtb* to the LC3-positive phagophore.

it binds to myosin VI indicates that TAX1BP1 may play a role in physical movement of large cargo such as bacteria or mitochondria perhaps to facilitate fusion with endosomes or the lysosome (*Figure 7*; *Tumbarello et al., 2012*). Alternatively, the recruitment of extant endosomal vesicles may represent a cellular mechanism that counteracts pathogen-mediated phagosomal disruptions by providing new material for membrane repair, perhaps working in concert with autophagy.

Although TAX1BP1 plays a role in cell intrinsic defense via autophagy, its complex domain structure and known interaction partners suggests it may have a broader role in antibacterial defense in the context of an intact immune system. Originally identified as an A20-binding protein (*Heyninck et al., 1999*), TAX1BP1 activates this ubiquitin editing enzyme to downregulate inflammatory responses by inhibiting TNF-α and IL-1β-induced NF-kB activation (*Iha et al., 2008*; *Shembade et al., 2011*). Given that hyperinflammatory responses promote *Mtb* infection (*Olive and Sassetti, 2018*), we reasoned TAX1BP1 may also function to limit these responses, especially since we identified increased TAX1BP1 phosphorylation at Ser-693, a substrate for IKKα, during infection. However, we did not detect significant changes in pro-inflammatory cytokine levels (IL-6, TNF-α, or IL-12) from infected *Tax1bp1*$^{-/-}$ macrophage supernatants, although the increased type I interferon response during *Mtb* infection of these cells indicates that TAX1BP1 may modulate inflammation during *Mtb* infection to some degree.

In summary, we have created a unique resource that catalogs the dynamic changes in the macrophage proteome during the first 24 hr of infection with *Mtb*. We envision that this information will catalyze future efforts to ultimately understand the full range of mechanisms by which macrophages respond to infection and carry out their pleiotropic functions. Indeed, from this work we identified of eight new PTMs within four autophagy receptors (BNIP3L, Optineurin, FUNDC1, and p62). By using our PTM dataset to inform genetic studies, we discovered that the unique TAX1BP1 receptor functions at a discrete step in autophagic targeting of bacilli to the

lysosome, and is required for limiting *Mtb* replication. Future comparative studies will shed light on responses that are unique to *Mtb* infection, and thus identify how this notorious pathogen perturbs or hijacks cellular responses for its own benefit.

# Materials and methods

## Key resources table

| Reagent type (species) or resource | Designation | Source or reference | Identifiers | Additional information |
|---|---|---|---|---|
| Cell line (*M. musculus*) | RAW 264.7 | ATCC | TIB-71 | |
| Transfected construct | pENTR-N-FLAG | *Parry* et al., in preparation | | Lentiviral construct to transfect and express FLAG-tagged autophagy receptors |
| Transfected construct | plentiGuide puro | Addgene | 52963 | Lentiviral construction to transfect and expressed guide RNAs |
| Biological sample (*M. musculus*) | Primary bone marrow-derived macrophages, WT C57BL/6J | Jackson Laboratory | Stock # 000664 | Freshly isolated from mice |
| Biological sample (*M. musculus*) | Primary bone marrow-dervied macrophages Rosa26-Cas9 knockin mouse | Jackson Laboratory (*Platt et al., 2014*) | Stock # 026179 | Freshly isolated from mice |
| Antibody | anti-beta-actin (Mouse monoclonal) | Santa Cruz | 47778 | WB (1:1000) |
| Antibody | Anti-FLAG M2 (Mouse monoclonal) | Sigma | F3165 | WB (1:1000) IF (1:500) |
| Antibody | anti-p62 (Mouse monoclonal) | Abcam | 109012 | 1 µg/ml |
| Antibody | Anti-phospho-Ser172-TBK1 (D52C2) | Cell signalling | 5483 | IF (1:400) |
| Antibody | Anti-Ubiquitin (FK2) | Millipore Sigma | 04–263 | IF (1:400) |
| Antibody | Anti-LC3 (clone 2G6) | Nanotools | 0260–100/LC3-2G6 | IF (1:200) |
| Sequence-based reagent | PCR primers | IDT | PCR primers | Please see *Supplementary file 2* |
| Commercial assay | Luciferase assay system | Promega | E1500 | |
| Commercial kit | Mouse IL-6 DuoSet ELISA | R and D | DY406 | |
| Commercial kit | Mouse TNF-α DuoSet ELISA | R and D | DY410 | |
| Commercial kit | Mouse IL-12/IL-23 p40 allele-specific DuoSet ELISA | R and D | DY499 | |
| Chemical compound, drug | Recombinant murine IFN-g | Peprotech | 315–05 | |
| Chemical compound, enzyme | Sequencing grade modified trypsin | Promega | V511 | |
| Software, algorithm | MaxQuant | (*Cox and Mann, 2008*) | | MS Database search software |
| Software, algorithm | artMS | (*Jimenez-Morales et al., 2019*) | | MS Statistical analysis software |
| Other | DAPI stain | Invitrogen | D1306 | (1 µg/mL) |

*Continued on next page*

*Continued*

| Reagent type (species) or resource | Designation | Source or reference | Identifiers | Additional information |
|---|---|---|---|---|
| Strain | *Mtb*: Erdman::pmv261Zeo_MCherry | (*Penn et al., 2018*) | | mCherry-expressing *Mtb* Erdman strain |
| Strain | *Mtb*: Erdman (expressing *luxCDABE* operon) | (*Roberts et al., 2019*) | | luciferase-expressing *Mtb* Erdman strain |
| Strain | *Mtb*: Erdman | ATCC | 35801 | |

## Ethics statement

An animal use protocol (AUP-2015-11-8096) for mouse use was approved by the Office of Laboratory and Animal Care at the University of California, Berkeley, in adherence with guidelines from the Guide for the Care and Use of Laboratory Animals of the National Institutes of Health.

## Mice and macrophages

Wild-type C57BL/6J mice obtained from Jackson laboratories. $p62^{-/-}$ mice were provided by Herbert Virgin at Washington University, St. Louis. $Tax1bp1^{-/-}$ mice were obtained from Hidekatsu Iha at the University of Oita, Japan. Rosa26-Cas9 knock-in mice (*Platt et al., 2014*) were provided by Gregory Barton at the University of California, Berkeley. Primary murine bone-marrow derived macrophages (BMDMs) were prepared by flushing femurs from 8- to 12 week-old male and female mice. Bone marrow extracts were differentiated for 7 days and cultured in DMEM, high glucose (Gibco) supplemented with 20% FBS, 2 mM glutamine, and 15% MCSF derived from 3T3-MCSF cells. Raw264.7 cells were obtained from the ATCC and cultured in DMEM, high glucose supplemented with 10% FBS, 20 mM HEPES, and 2 mM glutamine. Mycoplasma contamination was not detected. According to the ATCC at the time of manuscript preparation, short tandem repeat (STR) testing was under development and not available for mouse cell line validation.

## Bacterial strain

*M. tuberculosis* (Erdman) was used for macrophage infections. *M. tuberculosis* was grown to log phase in 7H9 liquid media (BD) supplemented with 10% Middlebrook OADC (Sigma), 0.5% glycerol, 0.05% Tween80 in roller bottles at 37°C.

## Macrophage infection with *M. tuberculosis*

For proteomics, three independent experiments were performed. BMDMs were seeded at a density of $3 \times 10^6$ cells per well in four-well cell culture treated dishes (Nunc OmniTray). Six dishes were seeded for each experimental condition. The cells were cultured at 37°C with 5% $CO_2$ for 72 hr prior to infection.

For microscopy to detect FLAG-tagged autophagy receptor colocalization, three coverslips for each experimental condition were seeded with RAW macrophages at a density of $1.2 \times 10^5$ cells per coverslip in a 24-well dish the day before infection. For microscopy to detect ubiquitin, p62, and phospho-TBK1, four wells for each experimental condition were seeded with bone-marrow-derived macrophages at a density of $3 \times 10^4$ cells per well in a 96-well plate. The cells were cultured for 18 hr prior to infection.

In order to synchronize the start time of the infections, macrophages were infected with *M. tuberculosis* (Erdman strain) using a 'spinfection protocol' at a multiplicity of infection of 10 for proteomics or two for microscopy experiments. Logarithmic phase *M. tuberculosis* suspensions were washed twice with PBS followed by centrifugation for 5 min at 1462 g and sonication to remove and disperse clumps, respectively. *M. tuberculosis* was resuspended in DMEM with 10% horse serum. Media was removed from the macrophage monolayers, the bacterial suspension was overlaid, and centrifugation was performed for 10 min at 162 *g*. For proteomics experiments, cells were returned to macrophage media and incubated until harvesting at 2-, 4-, 6-, 8-, or 24 hr post-infection. Uninfected mock controls were harvested at 0-, 6-, and 24 hr intervals. For microscopy experiments, infected monolayers were washed twice with PBS and then macrophage media was overlaid. At 8 hr post-

infection, the macrophages were washed once with PBS, fixed with 4% paraformaldehyde, washed twice with PBS, and immunostained.

## Cell lysis

At each time point 0–24 hr post-infection, macrophages were washed with PBS warmed to 37°C and *M. tuberculosis* was sterilized on the cell culture plates by the addition of 100% methanol at 4°C. The cells were washed twice with PBS and lysed with 6 ml of lysis buffer (8 M urea, 150 mM NaCl, 100 mM ammonium bicarbonate, pH 8; added per 10 ml of buffer: 1 tablet of Roche mini-complete protease inhibitor EDTA free and 1 tablet of Roche PhosSTOP tablet) prepared fresh before each replicate. Lysates were stored at −80°C until further processing.

## Trypsin digest and desalting

Lysates were sonicated three times with a Sonics VibraCell probe tip sonicator at 7 watts for 10 s. In order to remove insoluble precipitate, lysates were centrifuged at 16,100 g at 4°C for 30 min. A bicinchoninic acid assay (Pierce) was performed to measure protein concentration in cell lysate supernatants. 10 mg of each clarified lysate was reduced with 4 mM tris(2-carboxyethyl)phosphine for 30 min at room temperature and alkylated with 10 mM iodoacetamide for 30 min at room temperature in the dark. Remaining alkylated agent was quenched with 10 mM 1,4-dithiothreitol for 30 min at room temperature in the dark. The samples were diluted with three starting volumes of 100 mM ammonium bicarbonate, pH 8.0, to reduce the urea concentration. Samples were incubated with 100 µg of sequencing grade modified trypsin (Promega) and incubated at room temperature with rotation for 20 hr. The sample pH was reduced to approximately 2.0 by the addition of 10% trifluoroacetic acid (TFA) to a final concentration of 0.3% trifluoroacetic acid followed by 6 M HCl at a 1:100 vol ratio. Insoluble material was removed by centrifugation at 1989 g for 10 min.

Peptides were desalted using SepPak C18 solid-phase extraction cartridges (Waters). The columns were activated with 1 ml of 80% acetonitrile (ACN), 0.1% TFA, and equilibrated with 3 ml of 0.1% TFA. Peptide samples were applied to the columns, and the columns were washed with 3 ml of 0.1% TFA. Peptides were eluted with 1.1 ml of 40% ACN, 0.1% TFA. Peptides were divided for global protein analysis (10 µg), phosphoenrichment (1 mg), or diGly-enrichment (remaining sample).

## Global protein analysis

10 µg of peptides were dried with a centrifugal evaporator and stored at −20°C until analysis by liquid chromatograph and mass spectrometry.

## Phosphopeptide enrichment by immobilized metal affinity chromatography

Iron nitriloacetic acid (NTA) resin were prepared in-house by stripping metal ions from nickel nitroloacetic acid agarose resin (Qiagen) with 500 mM ethylenediaminetetraacetic acid, pH 8.0 three times. Resin was washed twice with water and 100 mM iron(III) chloride was applied three times. The iron-NTA resin was washed twice with water and once with 0.5% formic acid. Iron-NTA beads were resuspended in water to create a 25% resin slurry. 50 µl of Fe-NTA resin slurry was transferred to individual Silica C18 MicroSpin columns (The Nest Group) pre-equilibrated with 100 µl of 80% CAN, 0.1% TFA on a vacuum manifold. Subsequent steps were performed with the Fe-NTA resin loaded on the Silica C18 columns.

Peptide samples were mixed twice with the Fe-NTA resin and allowed to incubate for 2 min. The resin was rinsed four times with 200 µl of 80% ACN, 0.1% TFA. In order to equilibrate the chromatography columns, 200 µl of 0.5% formic acid was applied twice to the resin and columns. Peptides were eluted from the resin onto the C18 column by application of 200 µl of 500 mM potassium phosphate, pH 7.0. Peptides were washed twice with 200 µl of 0.5% formic acid. The C18 columns were removed from the vacuum manifold and eluted twice by centrifugation at 1000 g with 75 µl of 50% ACN, 0.1% TFA. Peptides were dried with a centrifugal adaptor and stored at −20°C until analysis by liquid chromatograph and mass spectrometry.

## Di-glycine peptide enrichment by immunoprecipitation

Peptide samples were subjected to ubiquitin remnant immunoaffinity purification (*Udeshi et al., 2013*) with 31 μg of ubiquitin remnant antibody (Cell Signaling). Peptides were lyophilized for 2 days to remove TFA in the elution. The lyophilized peptides were resuspended in 1 ml of IAP buffer (50 mM 4-morpholinepropnesulfonic acid, 10 mM disodium hydrogen phosphate, 50 mM sodium chloride, pH 7.2). Peptides were sonicated and centrifuged for 5 min at 16,100 g. Ubiquitin remnant beads were washed twice with IAP buffer and incubated with the peptides at 4°C for 90 min with rotation. Unbound peptides were separated from the beads after centrifugation at 700 g for 60 s. Beads containing peptides with di-glycine remnants were washed twice with 500 μl of water and peptides were eluted twice with 55 μl of 0.15% TFA. Di-glycine remnant peptides were desalted with UltraMicroSpin C18 column (The Nest Group). Desalted peptides were dried with a centrifugal adaptor and stored at −20°C until analysis by liquid chromatograph and mass spectrometry.

## Liquid chromatography and mass spectrometry

Peptides were analyzed in technical duplicate on a ThermoFisher Orbitrap Fusion Lumos Tribid mass spectrometry system equipped with an Easy nLC 1200 ultrahigh-pressure liquid chromatography system interfaced via a Nanospray Flex nanoelectrospray source. Samples were injected on a C18 reverse phase column (25 cm x 75 μm packed with ReprosilPur C18 AQ 1.9 um particles). Peptides were separated by an organic gradient from 5% to 30% ACN in 0.02% heptafluorobutyric acid over 180 min at a flow rate of 300 nl/min for the phosphorylated peptides or unmodified peptides for global abundance. Ubiquitylated peptides were separated by a shorter gradient of 140 min. Spectra were continuously acquired in a data-dependent manner throughout the gradient, acquiring a full scan in the Orbitrap (at 120,000 resolution with an AGC target of 400,000 and a maximum injection time of 50 ms) followed by as many MS/MS scans as could be acquired on the most abundant ions in 3 s in the dual linear ion trap (rapid scan type with an intensity threshold of 5000, HCD collision energy of 32%, AGC target of 10,000, maximum injection time of 30 ms, and isolation width of 0.7 *m/z*). Singly and unassigned charge states were rejected. Dynamic exclusion was enabled with a repeat count of 2, an exclusion duration of 20 s, and an exclusion mass width of ±10 ppm.

## Label-free quantitation and proteomics statistical analysis

Mass spectrometry data was assigned to mouse sequences and MS1 intensities were extracted with MaxQuant (version 1.6.0.16) (*Cox and Mann, 2008*). Data were searched against the SwissProt mouse protein database (downloaded on March 3, 2018). Trypsin (KR|P) was selected allowing for up to two missed cleavages. Variable modification was allowed for N-terminal protein acetylation and methionine oxidation in addition to phosphorylation of serine, threonine, and tyrosine, or diGly modification of lysine, as appropriate. A static modification was assigned to carbamidomethyl cysteine. The other MaxQuant settings were left at the default.

Statistical analysis of MaxQuant-analyzed data was performed with artMS Bioconductor package (*Jimenez-Morales et al., 2019*), which performs the relative quantification using the MSstats Bioconductor package (version 3.14.1) (*Choi et al., 2014*). artMS version 0.9 was used for statistical analysis. Contaminants and decoy hits were removed. The samples were normalized across fractions by median-centering the log$_2$-transformed MS1 intensity distributions. The MSstats group comparison function was run with no interaction terms for missing values, no interference, unequal intensity feature variance, restricted technical and biological scope of replication. Log$_2$(fold change) for protein/sites with missing values in one condition but found in $\geq2$ biological replicates of the other condition of any given comparison were estimated by imputing intensity values from the lowest observed MS1-intensity across samples (*Webb-Robertson et al., 2015*); p values were randomly assigned between 0.05 and 0.01 for illustration purposes. Statistically significant changes were selected by applying a log$_2$-fold-change ($>1.0$ or$<-1.0$) and a p value ($<0.05$). Statistical analysis of the phosphopeptide MS data (SwissProt mouse protein database downloaded on April 4, 2017) was also performed with the updated release of artMS version 1.4.0 and these results were incorporated into *Figures 1–3*, *Supplementary files 3* and *5*, the NDEx cell network, and TB Omics Explorer Web apps.

The mass spectrometry proteomics RAW mass spectrum files and all database searches have been deposited to the ProteomeXchange Consortium via the PRIDE (*Perez-Riverol et al., 2019*; *Deutsch et al., 2017*) partner repository with the dataset identifier PXD015361.

## Cell network map

The *Mtb*-human protein-protein interaction and statistically significant proteomics PTM data ($\log_2$(-fold change)>1 or $\leq$-1, p value < 0.05) were imported into Cytoscape version 3.7.2, and the modified cell network exported to NDEx (*Pratt et al., 2015*) for a viewable and searchable network on-line: http://ndexbio.org/#/network/2f38c264-1e05-11ea-bb65-0ac135e8bacf.

## Web app

Proteomics and RNAseq datasets were assembled into an on-line application, the TB Omics Explorer (https://artms.shinyapps.io/tb-omics-explorer/). The data is searchable by biological complex, gene, or protein name.

## Gene enrichment analysis

Gene enrichment analysis was performed with Metascape (metascape.org) (*Zhou et al., 2019*) with Custom analysis settings using only GO Biological Processes. Heatmaps were made with the R-based Complex heat map program and the Ward D clustering algorithm.

## PhosFate analysis

Murine phosphorylation sites were mapped to the human proteome using R. Phosphopeptide $\log_2$-fold-change profiles were uploaded to the PhosFate Profiler tool (Phosfate.com ; *Ochoa et al., 2016*) for kinase and kinase complex predictions.

## RNAseq

Bone-marrow-derived macrophages were infected at a MOI of 1 for the 24 hr time point and a MOI of 10 for 2- and 6 hr time points. At the indicated time points, monolayers were washed with PBS, and macrophages were resuspended in 1 ml of TRIzol (ThermoFisher). Following the addition of chloroform (200 μl), samples were mixed and centrifuged for 10 min. at 4°C. An equal volume of 70% ethanol was added to the aqueous layer, and RNA was extracted using silica spin columns (Invitrogen PureLink RNA Mini Kit). Purified RNA was treated DNAse (New England Biolabs) followed by EDTA. RNAseq libraries were prepared from biological triplicate samples by the University of California, Davis, Expression Analysis Core. Differential gene expression analysis was performed by the University of California, Davis, Bioinformatics Core. Transcriptional changes of similar levels between the 24 hr datasets (standard deviation <1 for the $\log_2$fold changes) measured in our laboratory and a previously published study of *Mtb* infected bone derived macrophages were included in the analysis.

## Cloning and lentiviral transduction of FLAG-tagged autophagy receptors

PCR products were generated for each autophagy receptor using the following primer pairs: P30/P2 (p62), P31/P4 (BNIP3), P32/P6 (BNIP3L), P33/P8 (FUNDC1; *Supplementary file 2*). The remaining autophagy receptors were cloned by splicing with overlap extension PCR with the following primers: P34/P37 and P38/P10 (NBR1), P35/P39 and P40/P12 (Optineurin), P36/P41 and P42/P14 (TAX1BP1). PCR products were cloned into an entry vector encoding a N-terminal triple FLAG tag (pENTR-N-FLAG; Parry et al., in preparation) using sequence- and ligation-independent cloning. The Gateway LR reaction was used to introduce the cloned autophagy receptors into the pLenti CMV Puro DEST destination vector (*Campeau et al., 2009*). RAW264.7 cells were transduced and subjected to puromycin selection for 7 days.

## Cloning and lentiviral transduction of CRISPR guide RNAs

Single oligonucleotide guides for three scramble controls and two exons in *Tax1bp1* selected from the Brie library were closed into lentiGuide puro (Addgene #52963) using the following primer pairs: P174/P175 (Scramble 1), P176/P177 (Scramble 2), P178/P179 (Scramble 3), P160/P161 (*Tax1bp1*

exon 10), and P164/P165 (*Tax1bp1* exon 9). 293 T cells were reverse transfected with the vectors encoding scramble or *Tax1bp1* guides. Bone marrow cells from Rosa26-Cas9 knock-in mice were transduced with retrovirus and differentiated in the presence of puromycin. Transduced and fully-differentiated bone-marrow-derived macrophages were cultured for 3 days in the absence of puromycin before seeding for microscopy experiments.

## Western blot

The anti-FLAG M2 mouse monoclonal antibody was purchased from Sigma-Aldrich. 60 μg of protein lysate was separated by SDS-PAGE (BioRad Miniprotean TGX 4–20%) and transferred onto nitrocellulose membranes. After blocking with Odyssey blocking buffer and probing with the anti-FLAG M2 antibody at a dilution of 1:1000, membranes were imaged on an Odyssey scanner (Li-cor).

In order to probe for actin on the same nitrocellulose membrane, antibodies were stripped from the membrane with 0.2 N sodium hydroxide and re-probed with the ß-actin mouse monoclonal antibody (C4; Santa Cruz Biotechnology, SC-47778).

## Immunofluorescence microscopy for FLAG-tagged autophagy receptor colocalization

Fixed infected macrophages were washed with PBS and incubated with blocking and permeabilization buffer (0.05% saponin, 5% fetal calf serum, PBS) for 30 min at room temperature. Coverslips were incubated with anti-FLAG M2 mouse monoclonal antibody at a dilution of 1:500 for 3 hr at room temperature and 1 hr with secondary antibody conjugated to fluorophore at a dilution of 1:1000 (Invitrogen). 45 images per coverslip were obtained with a Nikon Eclipse TE2000-E spinning disc confocal microscope equipped with an Andor laser system and Borealis beam conditioning unit. Images were obtained with a 100X oil objective for figures and 40X air objective for colocalization.

## Immunofluorescence microscopy for p62, ubiquitin, and phospho-TBK1

Immunostaining for p62 and ubiquitin was performed with blocking and permeabilization buffer (1% saponin, 3% bovine serum albumin) and anti-p62 rabbit monoclonal antibodies (Abcam #AB109012) at a concentration of 1 μg/ml or anti-ubiquitylated protein antibodies (clone FK2, Millipore Sigma #04–263) at a dilution of 1:400 for 3 hr at room temperature. Immunostaining for phospho-TBK1 was performed with blocking and permeabilization buffer (0.3% Triton X-100, 1% bovine serum albumin in PBS) and anti-phospho-Ser172-TBK1 (D52C2) XP Rabbit monoclonal antibody (Cell Signaling #5483) for three hours at room temperature. Samples were incubated with a secondary antibody conjugated to Alexa Fluor 488 fluorophore at a dilution of 1:4000 (Invitrogen) and DAPI at a dilution of 1:1000 for 1 hr at room temperature. 25 images per well were obtained with a Perkin Elmer Opera Phenix High Content Screening confocal microscope at 40X magnification for colocalization or 63X for figures. Automated colocalization measurements were performed with the Perkin Elmer Harmony software package. A pipeline was created to measure colocalization of *Mtb* and autophagy markers (p62, FK2, and phospho-TBK1) in macrophages infected with two *Mtb* bacilli.

## Immunofluorescence microscopy for LC3

Immunostaining for LC3 was performed with blocking and permeabilization buffer (0.3% Triton X-100, 2% bovine serum albumin) and anti-LC3 (clone 2G6, Nanotools #0260–100/LC3-2G6) at a dilution of 1:200 for 3 hr at room temperature. Samples were incubated with a secondary antibody conjugated to Alexa Fluor 647 fluorophore at a dilution of 1:1000 (ThermoFisher) and DAPI at a dilution of 1:1000 for one hour at room temperature. 59 images per well were obtained with the Opera Phenix confocal microscope at 40X magnification for colocalization or 63X for figures. Automated colocalization measurements were performed in macrophages infected with two *Mtb* bacilli.

## *Mtb* growth assays

Wild-type *Mtb* or *Mtb* transformed with the *luxCDABE* operon were prepared and inoculated as previously described (*Penn et al., 2018*; *Roberts et al., 2019*) in 96-well plates, and media was changed daily for macrophage cultures. To ensure equivalent plating numbers of wild-type and *Tax1bp1*$^{-/-}$ macrophages on day 0 of infection, separate 96-well plates containing the same cell dilutions plated in parallel were fixed with 4% PFA, nuclei were stained with DAPI, and the nuclei

number in each well was counted at 5X magnification with the Opera Phenix. To stimulate macrophages with IFN-γ, media was supplemented with 1.5 ng/μl of murine IFN- γ (Peprotech) starting at 18 hr prior to infection and fresh IFN-γ was supplemented in the media daily. Luminescence in *Mtb*-infected macrophages was measured as previously described (*Penn et al., 2018*).

For *Mtb* CFU experiments, macrophages were plated into 96-well plates with $3 \times 10^4$ cells per well and allowed to adhere for 24 hr prior to infection. *Mtb* CFU enumeration was performed as previously described (*Sogi et al., 2017*).

### Nitrite and cytokine measurements

Macrophages were infected at a M.O.I. of 2 or 10 and 24 hr post-infection the supernatants were collected for measurement of nitrite or cytokines, respectively. Nitrite was measured by the Griess reaction as described previously (*Roberts et al., 2019*). Type I interferon activity was measured in the supernatants using L929 ISRE-luciferase reporter cells as previously described (*Watson et al., 2015*). Mouse IL-12/IL-23 p40, IL-6, and TNF-α levels were measured in the supernatants using Duo-Set ELISA kits (R and D systems) per the manufacturer's instructions.

### Statistics

Statistical analysis of data was performed using GraphPad Prism eight software. Two-tailed unpaired Student's t-tests were used for analysis of microscopy images, *Mtb* growth assays, and cytokine measurements. Exact p values are listed in *Supplementary file 7*.

## Acknowledgements

We acknowledge members of the Cox and Krogan labs for their comments on this manuscript submission. We thank H Skip Virgin (Washington University, St. Louis, USA) and Hidekatsu Iha (Oita University, Japan) for $p62^{-/-}$ and $Tax1bp1^{-/-}$ mice, respectively. We thank Lori Kohlstaedt at the UC Berkeley proteomics core for assistance in LC-MS analysis of our peptide samples, and Christopher Noel for assistance in the development of the Perkin Elmer Harmony microscopy colocalization pipelines. This work was supported by NIH grants P01 AI063302 (NJK, JSC), P50 GM082250 (NJK), U19 AI106754 (JSC, NJK), DP1 AI124619 (JSC), and R01 AI120694 (NJK and JSC). JMB was supported by a NIH T32 training grant (4T32HL007185-39 and −40), K12 (5K12HL119997-05), K08 (1K08AI146267-0), and Cystic Fibrosis Foundation Harry Shwachman Award. Research reported in this publication was supported by the Office of The Director, National Institutes of Health of the National Institutes of Health under Award Number S10OD021828 for the Opera Phenix and associated staff support. The content is solely the responsibility of the authors and does not necessarily represent the official views of the National Institutes of Health.

## Additional information

### Funding

| Funder | Grant reference number | Author |
| --- | --- | --- |
| National Institute of Allergy and Infectious Diseases | P01 AI063302 | Jeffery S Cox<br>Nevan J Krogan |
| National Institute of General Medical Sciences | P50 GM082250 | Nevan J Krogan |
| National Institute of Allergy and Infectious Diseases | U19 AI106754 | Nevan J Krogan<br>Jeffery S Cox |
| National Institute of Allergy and Infectious Diseases | DP1 AI124619 | Jeffery S Cox |
| National Institute of Allergy and Infectious Diseases | R01 AI120694 | Jeffery S Cox<br>Nevan J Krogan |
| National Institute of Allergy and Infectious Diseases | 1K08AI146267 | Jonathan M Budzik |
| Cystic Fibrosis Foundation | Harry Shwachman Award | Jonathan M Budzik |

The funders had no role in study design, data collection and interpretation, or the decision to submit the work for publication.

**Author contributions**
Jonathan M Budzik, Conceptualization, Data curation, Formal analysis, Funding acquisition, Investigation, Visualization, Methodology; Danielle L Swaney, Resources, Data curation, Software, Methodology; David Jimenez-Morales, Data curation, Software, Formal analysis; Jeffrey R Johnson, Data curation, Software, Methodology; Nicholas E Garelis, Data curation, Formal analysis, Visualization; Teresa Repasy, Investigation, Methodology; Allison W Roberts, Lauren M Popov, Trevor J Parry, Investigation; Dexter Pratt, Resources, Software, Formal analysis; Trey Ideker, Resources, Software; Nevan J Krogan, Conceptualization, Resources, Software, Funding acquisition, Methodology; Jeffery S Cox, Conceptualization, Resources, Supervision, Funding acquisition, Investigation, Methodology, Project administration

**Author ORCIDs**
Jonathan M Budzik (iD) https://orcid.org/0000-0001-8025-7911
Danielle L Swaney (iD) http://orcid.org/0000-0001-6119-6084
Allison W Roberts (iD) http://orcid.org/0000-0001-6681-4144
Jeffery S Cox (iD) https://orcid.org/0000-0002-5061-6618

**Ethics**
Animal experimentation: An animal use protocol (AUP-2015-11-8096) for mouse use was approved by the Office of Laboratory and Animal Care at the University of California, Berkeley, in adherence with guidelines from the Guide for the Care and Use of Laboratory Animals of the National Institutes of Health.

**Decision letter and Author response**
Decision letter https://doi.org/10.7554/eLife.51461.sa1
Author response https://doi.org/10.7554/eLife.51461.sa2

# Additional files

**Supplementary files**
• Supplementary file 1. Rho values for peptide intensity levels from biological and technical replicates.

• Supplementary file 2. Primers used in this study.

• Supplementary file 3. Autophagy receptors changing in protein abundance, phosphorylation, or ubiquitylation during *Mtb* infection.

• Supplementary file 4. Protein abundance changes during *Mtb* infection.

• Supplementary file 5. Changes in phosphorylation during *Mtb* infection.

• Supplementary file 6. Changes in ubiquitylation during *Mtb* infection.

• Supplementary file 7. p values for *Figure 5*, *Figure 5—figure supplement 1*, *Figure 6*, *Figure 6—figure supplement 1*, *Figure 6—figure supplement 2*.

• Transparent reporting form

## Data availability

The mass spectrometry proteomics data have been deposited to the ProteomeXchange Consortium via the PRIDE partner repository with the dataset identifier PXD015361. Source microscopy files, co-localisation data and measurements of fold-change have been uploaded to Dryad.

The following datasets were generated:

| Author(s) | Year | Dataset title | Dataset URL | Database and Identifier |
| --- | --- | --- | --- | --- |
| Budzik JM, Cox JSC | 2019 | Global proteomic profiling of primary macrophages during M. tuberculosis infection identifies TAX1BP1 as a mediator of autophagy targeting | https://www.ebi.ac.uk/pride/archive/projects/PXD015361 | PRIDE, PXD015361 |
| Jonathan M Budzik, Danielle L Swaney, David Jimenez-Morales, Jeffrey R Johnson, Nicholas E Garelis, Teresa Repasy, Allison W Roberts, Lauren M Popov, Trevor J Parry, Dexter Pratt, Trey Ideker, Nevan J Krogan, Jeffery S Cox | 2020 | Dynamic post-translational modification profiling of M. tuberculosis-infected primary macrophages | https://doi.org/10.7272/Q6JQ0Z6J | Dryad Digital Repository, 10.7272/Q6JQ0Z6J |

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
