## [Decision Letter]

**Acceptance summary:**

Your study represents a comprehensive analysis that is distinguished from previous such work in its inclusion of quantitative proteomics of, not only abundance, but post-translational modifications (phosphorylation and ubiquitinylation) over a time course during *Mycobacterium tuberculosis* colonization of macrophages. A number of intriguing results were noted such as effects on DNA damage repair, actin cytoskeleton, and cell replication. The use of phosphorylation data to make predictions about host kinases that may have important roles in responding to *M. tuberculosis* infection represents a noteworthy advance. This proteomic dataset will be a valuable resource for tuberculosis researchers with regards to ongoing studies and it should also be valuable as a hypothesis generating resource.

**Decision letter after peer review:**

Thank you for submitting your article "Dynamic PTM profiling of *M. tuberculosis*-infected macrophages identifies TAX1BP1 as a critical mediator of xenophagy" for consideration by *eLife*. Your article has been reviewed by three peer reviewers, and the evaluation has been overseen by a Reviewing Editor and Wendy Garrett as the Senior Editor. The following individual involved in review of your submission has agreed to reveal their identity: Neeraj Dhar (Reviewer #3).

The reviewers have discussed the reviews with one another and the Reviewing Editor has drafted this decision to help you prepare a revised submission.

Summary:

The manuscript presents a global quantitative proteomic analysis of host proteins of *M. tuberculosis* infected macrophages. It provides a comprehensive analysis, that is distinguished from prior work, by its inclusion of quantitative proteomics of both abundance and phosphorylation or ubiquitinylation over a time course. This proteomic dataset represents a valuable resource for *M. tuberculosis* researchers with regard to ongoing studies and it should also be valuable as a hypothesis generating resource. A functional pathway enrichment analysis revealed cellular trafficking, metabolism, signal transduction, nucleocytoplasmic transport and response to DNA damage as statistically significantly enriched pathways. Changes in protein abundance revealed enrichment of antiviral and inflammatory pathways in infected macrophages. Another observation made in this study was that there was a statistically significant increase in the autophagy receptors p62 and TAX1BP1 during *M. tuberculosis* infection and that p62, NBR1, optineurin, and TAX1BP1 colocalized with *M. tuberculosis* during macrophage infection. The authors then chose to further investigate the functional role of Tax1BP1 using Tax1bp1-/- cells and a genetic approach. They found that TAX1BP1 deficiency resulted in the accumulation of ubiquitylated *M. tuberculosis* and resulted in increased intracellular bacterial replication. The reviewers have expressed general enthusiasm for the work but there are some concerns that need to be addressed.

Essential revisions:

1) TAX1BP1, which is already known to be an autophagy receptor, is the only protein investigated further. The downstream analysis of TAX1BP1 does not probe the significance of its phosphorylation in response to *M. tuberculosis* infection, which appears to be the logical course of action to follow given your findings. Moreover, the Discussion mentions additional possible functions for TAX1BP1 related to inflammatory cytokine responses, yet these were not investigated. Thus, it is not clear if autophagy or another function of TAX1BP1 is responsible for the modest CFU effects observed. Given that your submission represents an important technical and data resource to the field, we suggest that you carefully consider the data you have available for TAX1BP1 and present a more appropriate, toned down, conclusion. Also, as you have not conducted an extensive mechanistic investigation of these effects, it seems appropriate that you revised your title appropriately, removing TAX1BP1.

2) The co-localization with LC3 is not done in the presence of IFNγ. Considering that the CFU difference was abrogated in the presence of IFNγ, it would be valuable to determine if the LC3 co-localization findings change in the presence of IFNγ. This is not difficult to do and should be added to complete the data set.

---

## [Author Response]

Essential Revisions:1) TAX1BP1, which is already known to be an autophagy receptor, is the only protein investigated further. The downstream analysis of TAX1BP1 does not probe the significance of its phosphorylation in response to M. tuberculosis infection, which appears to be the logical course of action to follow given your findings. Moreover, the Discussion mentions additional possible functions for TAX1BP1 related to inflammatory cytokine responses, yet these were not investigated. Thus, it is not clear if autophagy or another function of TAX1BP1 is responsible for the modest CFU effects observed. Given that your submission represents an important technical and data resource to the field, we suggest that you carefully consider the data you have available for TAX1BP1 and present a more appropriate, toned down, conclusion. Also, as you have not conducted an extensive mechanistic investigation of these effects, it seems appropriate that you revised your title appropriately, removing TAX1BP1.

We agree with this comment and changed the title of the manuscript. The discussion of inflammation in the Discussion section was prompted primarily from the known role of TAX1BP1 in inflammatory responses in other contexts, and was speculative based on the exiting literature on the role of TAX1BP1 in mediating NF-κB inflammation^1^ the link between hyperinflammation and Mtb infection^2^. Indeed we did preliminary cytokine analysis and found that with IL-12, IL-6, and TNF-α there was no significant difference. However, we did see significantly more type I IFN. This data is presented is Figure 6—figure supplement 2 and discussed in the text of the manuscript (Discussion, first paragraph). We have toned down the role of inflammation in our Discussion and agree that it might be too speculative at this time.

2) The co-localization with LC3 is not done in the presence of IFNγ. Considering that the CFU difference was abrogated in the presence of IFNγ, it would be valuable to determine if the LC3 co-localization findings change in the presence of IFNγ. This is not difficult to do and should be added to complete the data set.

To complete the dataset, we performed the exact experiment requested and the results are now shown in Figure 6—figure supplement 1. In summary, the effects of IFN-γ on bacterial growth was mirrored by the changes in LC3 colocalization. We suspect this is due to high levels of nitric oxide, one of the most potent antimicrobial mechanisms active against Mtb, thereby affecting the kinetics, duration, or total amount of autophagy because the microbes are already attenuated.

References

1) Parvatiyar K, Barber GN, Harhaj EW. TAX1BP1 and A20 inhibit antiviral signaling by targeting TBK1-IKKi kinases. J Biol Chem. 2010 May 14; 285(20): 14999-15009. PMID: 20304918 PMCID: PMC2865285.

2) Moreira-Teixeira L, Mayer-Barber K, Sher A, O'Garra A. Type I interferons in tuberculosis: Foe and occasionally friend. J Exp Med. 2018 May 7; 215(5): 1273-1285. PMID: 29666166 PMCID: PMC5940272.